# Deep Discrete-Time Survival Analysis with Guaranteed Monotonicity

## Abstract

Discrete-time neural survival models trained with binary cross-entropy are attractive due to their simplicity. However, they can produce invalid patient-specific survival curves that increase over time when survival probabilities at different time points are learned without structural constraints. We propose Kaplan–Meier Net (KMNet), a discrete-time neural survival model that predicts interval-wise conditional survival probabilities and constructs the survival curve through a Kaplan–Meier style product, guaranteeing non-increasing survival predictions by design. KMNet is trained with a censoring-aware weighted binary cross-entropy objective and is further augmented with a smooth ranking term that compares individuals using the conditional survival probability at the event interval of the anchor observation, which differs from the global ranking losses used in existing deep survival models. We evaluate KMNet on eight benchmark datasets and compare it with seven strong neural baselines. Across eight datasets, KMNet attains the best average rank for both time-dependent concordance and integrated Brier score, without being uniformly superior on every dataset further D-calibration is not rejected on seven datasets, and all predicted survival curves are non-increasing by construction. [1]

## 1 Introduction

Survival analysis studies when an event happens, such as death, relapse, or re-incarceration, while accounting for the fact that some individuals may leave a study early or the event may not be observed within follow-up (right censoring). In many medical applications, the central goal is patient-specific prognosis. Given a patient's covariates $x$, we would like to estimate a survival curve $S(t \mid x)$, the probability of remaining event-free beyond time $t$. Such individualized survival curves summarize patient specific event risk over clinically relevant horizons and can support prognosis only insofar as they are trustworthy. This asks more than producing a plausible curve: at each horizon, the predicted risks should agree with the event frequencies actually observed, neither systematically over- nor understating risk across patients, and separating higher-from lower-risk individuals by a faithful, rather than exaggerated or overly cautious, margin. Even predictions that meet these conditions need not translate into clinical benefit, which further depends on how they are used to guide decisions.

In recent years, deep learning models have been increasingly used for time-to-event prediction when flexible function approximation is desired. A common practical choice is discrete-time modeling, where follow-up is represented on a fixed time grid, and the learning task becomes predicting survival behaviour one interval at a time. This setup integrates naturally with neural networks and has motivated a wide range of deep survival models (Katzman et al., 2018; Lee et al., 2018; Gensheimer & Narasimhan, 2019; Fotso, 2018; Yu et al., 2011). However, a fundamental requirement of any survival curve is that it should never increase over time. As time passes, the probability of having survived beyond that time can only stay the same or decrease. In widely used binary cross-entropy (BCE) based discrete-time model BCESurv (Kvamme & Borgan, 2019), survival probabilities at different grid points are often learned without an explicit cross-time

---

[1]Generative AI tools were used to assist with language editing and manuscript review. All technical content, analyses, results, and claims were independently verified by the authors, who take full responsibility for the manuscript.

constraint. As a result, the predicted curve can violate monotonicity, producing $\widehat{S}(t + \Delta \mid x) > \widehat{S}(t \mid x)$ for some $\Delta \geq 0$. This behaviour is incompatible with the definition of a survival function and can occur even when ranking-based metrics remain strong.

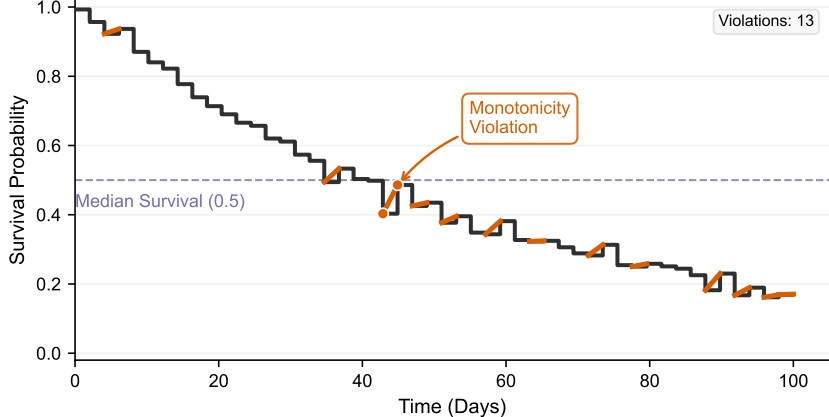 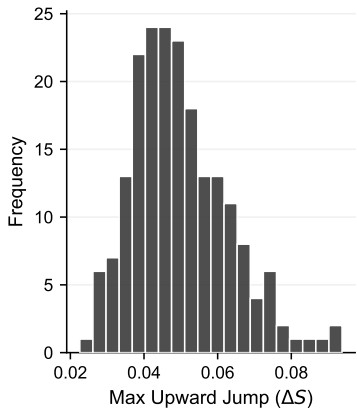

(a) Example BCESurv prediction for a single patient. The predicted survival curve should be non-increasing, but BCESurv may produce upward steps (highlighted), violating monotonicity and potentially creating ambiguous crossings of the 0.5 level used to define the median survival time.

(b) Distribution of the maximum upward jump $\Delta S = \max_k \max\left(0, \hat{S}(t_{k+1}) - \hat{S}(t_k)\right)$ across patients, quantifying the severity of monotonicity violations.

Figure 1: Motivation for KMNet: ~~BCE-based discrete-time survival models that~~ BCESurv directly predict survival at each time point can yield non-monotone survival curves, which contradicts the interpretation of survival probability and complicates downstream summaries such as the median survival time.

This behaviour of BCESurv is illustrated in Figure 1. In Subfigure 1a, we show the predicted survival curve for a representative patient from the Metabric study, which includes genomic and transcriptomic profiles from approximately 2,500 primary breast cancers, revealing 10 molecular subtypes with associated clinical outcomes (Curtis et al., 2012). The curve is not monotone and therefore violates a fundamental requirement of survival functions. Beyond being theoretically invalid, such non-monotonicity has direct practical consequences for interpretation and the generation of downstream clinical summaries. For instance, the median survival time is defined as the earliest time at which the survival probability falls below 0.5. When the predicted curve oscillates, the level 0.5 may be crossed multiple times (as in this example) or not crossed at all, and the resulting estimate becomes ambiguous or highly sensitive to small perturbations, unless additional ad hoc post-processing is applied. Subfigure 1b quantifies the magnitude of these monotonicity violations by reporting the empirical distribution of upward jump heights, highlighting that a non-negligible fraction of predictions exhibit substantial increases over time.

We address these limitations of BCESurv by adopting the conditional-survival construction used in Nnet (Gensheimer & Narasimhan, 2019), which directly parameterizes interval-specific conditional survival probabilities, and by augmenting it with an event-interval conditional ranking objective. We refer to the resulting discrete-time neural survival model as the Kaplan–Meier Net (KMNet). KMNet predicts per-interval conditional survival probabilities on a user-defined time grid and constructs individual survival functions through their cumulative product. As in established discrete-hazard models, this product-of-conditionals construction guarantees non-increasing survival curves by design while retaining the representational flexibility of neural networks. To further improve risk stratification, KMNet employs a smooth ranking objective that operates on the conditional survival probability at the event interval of the anchor individual, rather than on a global risk score or cumulative event probability as in DeepHit-style ranking. The contributions of this work are as follows:

1. We introduce KMNet, a discrete-time neural survival model that adopts a product-of-conditionals parameterization to generate patient-specific, non-increasing survival curves by construction, together with an event-interval conditional ranking objective that encourages improved risk discrimination.

2. We conduct an extensive empirical evaluation across eight benchmark survival datasets against seven established neural baselines, assessing discrimination, integrated prediction error, D-calibration, horizon-specific calibration, median-survival accuracy, and risk stratification. We further perform statistical significance tests and comparisons with monotone post-processed variants of BCESurv.

3. **We provide an open-source Python implementation of KMNet with a streamlined interface to support reproducibility and facilitate adoption by the research community. To preserve double-blind review, the PyPI link is withheld during review; however, the package materials are included in the supplementary material to support reviewers' evaluation.**

4. We conduct a controlled simulation study under a known data-generating mechanism to examine the sensitivity of KMNet to its loss weighting and time-discretization choices, and to illustrate their effects on discrimination, integrated Brier score, and risk stratification.

5. We perform controlled ablation studies to isolate the contribution of the proposed event-interval ranking objective and to assess the effects of alternative base losses, cumulative ranking formulations, ranking scales, and pairwise penalties while keeping the model architecture and evaluation protocol fixed.

The remainder of the paper is organized as follows. Section 2 reviews patient-specific survival curves, the Kaplan–Meier estimator, and discrete-time survival models trained with binary cross-entropy. Section 3 then introduces KMNet and its training objective. Section 4 presents the real-data evaluation, including dataset details, baseline methods, the experimental protocol, and comparative results. Section 5 presents two complementary ablation studies that isolate the proposed ranking component and examine the effects of alternative objective-design choices. Statistical significance analyses are reported in Section 6. Section 7 provides a controlled simulation study examining the sensitivity of KMNet to its loss weighting and time-discretization choices. Finally, Section 8 concludes with a discussion and directions for future work.

**Statement of Significance.** A limitation of marginal-survival BCE formulations such as BCESurv is that widely used binary cross-entropy-based models can produce non-monotone patient-specific survival curves, which are theoretically invalid and can undermine clinical interpretation. ~~Existing deep survival methods often achieve competitive discrimination, but they do not always guarantee survival predictions that remain non-increasing over time.~~ This paper introduces Kaplan–Meier Net (KMNet), a discrete-time neural survival model that predicts interval-wise conditional survival probabilities and constructs survival curves via a Kaplan–Meier-style product, thereby guaranteeing monotonicity by design. Across eight benchmark datasets, KMNet demonstrates ~~strong discrimination and calibration~~ competitive discrimination and generally favourable calibration performance, while consistently producing valid survival curves. This work may benefit researchers developing survival models and may support the interpretation of patient-specific survival predictions when appropriately validated. ~~for prognosis and decision-making.~~ Clinical use, however, would require prospective validation and assessment of decision utility in the intended application setting.

## 2 Background

In this section, we first introduce patient-specific survival functions and related notation. We then review the classical Kaplan–Meier estimator, which provides a population-level survival curve and does not incorporate patient covariates. Finally, we discuss discrete-time neural survival models trained with binary cross-entropy objectives and highlight the challenges that arise when translating their outputs into valid survival functions.

## 2.1   Patient-specific survival probability

Let $X \in \mathbb{R}^d$ denote a random covariate vector and let $T \geq 0$ denote the actual event time. ~~In a typical time-to-event studies, event times may be subject to right censoring.~~ In time-to-event studies, the event time may be only partially observed when follow-up ends before the event occurs, resulting in right censoring. Let $C \geq 0$ denote the censoring time and define the observed time and event indicator as

$$Y = \min(T, C), \qquad \delta = 1\{T \leq C\}.$$

Hence, $\delta = 1$ indicates that the event is observed at time $Y$, whereas $\delta = 0$ indicates that the observation is censored at time $Y$ and the event time is only known to satisfy $T > Y$. For a covariate value $x$, the patient-specific survival function[2] is defined as

$$S(t \mid x) = \Pr(T > t \mid X = x), \qquad t \geq 0.$$

The function $S(\cdot \mid x)$ gives the probability that a patient with covariates $x$ survives beyond time $t$. It satisfies $S(0 \mid x) = 1$, takes values in $[0, 1]$, and is non-increasing in $t$.

## 2.2   Kaplan–Meier survival probability

Kaplan-Meier survival probability estimation is a nonparametric method. Consider a cohort of $N$ individuals with observed time and event indicator $(Y_i, \delta_i)$ for $i = 1, \ldots, N$. The classical Kaplan–Meier estimator does not use covariates and targets the marginal survival function $S(t) = \Pr(T > t)$ under right censoring. Let $\tau_1 < \tau_2 < \cdots < \tau_K$ denote the distinct observed event times in the sample, i.e., the distinct values among $\{Y_i : \delta_i = 1\}$. For each event time $\tau_k$, define the risk set size and the number of events as

$$n_k = \sum_{i=1}^N 1\{Y_i \geq \tau_k\}, \qquad d_k = \sum_{i=1}^N 1\{Y_i = \tau_k, \ \delta_i = 1\}.$$

Here, $n_k$ counts the individuals still at risk at time $\tau_k$, and $d_k$ counts the events occurring at $\tau_k$. The The Kaplan–Meier estimator is defined as a product over event times,

$$\widehat{S}_{KM}(t) = \prod_{k: \ \tau_k \leq t} \left(1 - \frac{d_k}{n_k}\right), \qquad t \geq 0.$$

Each factor $\left(1 - \frac{d_k}{n_k}\right)$ lies in $[0, 1]$, so $\widehat{S}_{KM}(t)$ is non-increasing in $t$. Intuitively, $\frac{d_k}{n_k}$ estimates the conditional probability of experiencing the event at time $\tau_k$ among those at risk at $\tau_k$, Moreover, the Kaplan–Meier estimator combines these conditional quantities multiplicatively to obtain a survival curve.

In discrete-time settings, the same idea can be expressed on a fixed grid $0 = t_0 < t_1 < \cdots < t_J$ by defining a conditional survival probability at each grid point and then taking a product across time. This product construction is the key mechanism used later to ensure that predicted survival curves remain non-increasing.

## 2.3   Binary cross-entropy based discrete-time survival models

A common approach to discrete-time survival prediction converts time-to-event data into a sequence of binary classification problems on a fixed, researcher-specified grid $0 = t_0 < t_1 < \cdots < t_J$ over a time interval. Practitioners often use equally spaced grids, but this is not required and the grid may have non-equal spacing. For each individual $i$ and grid point $t_j$, a binary label is formed, typically $y_{ij} = 1\{Y_i > t_j\}$. A neural network is trained to output marginal survival probabilities $\widehat{S}(t_j \mid X_i) \in (0, 1)$ by minimizing a

---

[2]Terminology varies in the literature: some authors use the term *conditional survival* to mean conditioning on covariates, i.e., $S(t \mid x)$. In this work, we reserve *conditional survival probability* for the discrete-time setting, where survival is conditioned on having survived up to the previous grid point (e.g., $\Pr(T > t_j \mid T > t_{j-1}, X = x)$). Accordingly, we use *patient-specific survival probability* for covariate-conditioned survival $S(t \mid x)$, *patient-specific conditional survival probability* when conditioning on both covariates and prior survival, and omit "patient-specific" when covariates are not conditioned on.

weighted binary cross-entropy that accounts for right censoring by including only time points at which the individual is still observed and at risk. ~~This family of models is often referred to as BCE-based survival modeling, and BCESurv is a representative example that~~ BCESurv directly learns $\widehat{S}(t_j \mid x)$ at each grid point using a sigmoid output and binary cross-entropy loss.

While ~~these methods are~~ BCESurv is simple and effective for learning ~~risk scores~~ patient specific survival curve, they do not enforce the defining structural constraint of a survival function, namely, monotonicity in time. Because $\widehat{S}(t_j \mid x)$ is learned independently across $j$, it is possible to obtain non-monotone predictions with $\widehat{S}(t_{j+1} \mid x) > \widehat{S}(t_j \mid x)$ for some $j$, which is not a valid survival curve. This lack of structure also complicates the extraction of clinically meaningful summaries such as the median survival time. In a valid survival model, the median survival time is defined as $\inf\{t : S(t \mid x) \leq 0.5\}$ and can be read off as the first time the survival curve crosses 0.5. When the predicted curve is non-monotone, the crossing may not exist, may occur multiple times, or may be highly sensitive to small perturbations, making the median survival time ill-defined or unstable without additional post-processing such as isotonic regression or ad hoc smoothing. Consequently, ~~BCE-based~~ BCESurv models can produce probabilities that are difficult to interpret as proper patient-specific survival functions, even when their ranking performance is competitive.

## 2.4 Relation to Monotone Discrete-Time Survival Models

A valid survival curve must be non-increasing over time. Existing discrete-time survival models enforce this requirement usually through two types of parameterization: conditional hazards and event-time probability masses. Conditional-hazard models construct survival by multiplying the probabilities of surviving successive intervals, whereas probability-mass models construct survival by summing the probability assigned to event times beyond the current horizon. Both approaches guarantee monotonicity because they impose a coherent probabilistic structure across time. This distinguishes them from methods that predict marginal survival probabilities separately at each grid point, for which monotonicity is not automatically guaranteed.

Nnet models the probability of experiencing the event within each interval conditional on having survived to the beginning of that interval (Gensheimer & Narasimhan, 2019; Kvamme et al., 2019). Equivalently, the model may be viewed as predicting an interval-specific conditional survival probability. These quantities are constrained to lie between zero and one, and the survival curve is obtained by multiplying the conditional survival probabilities over successive intervals. Each additional time interval therefore multiplies the preceding survival probability by a number no greater than one, ensuring a non-increasing curve. The model is trained using the discrete-time survival likelihood, with uncensored observations contributing survival up to the event interval and failure within that interval, and censored observations contributing only over intervals known to have been survived.

An alternative family directly predicts a probability mass function over the possible event-time intervals (Kvamme & Borgan, 2021). A softmax-type normalization ensures that the interval probabilities are non-negative and sum to one. Survival at a particular horizon is then the total probability assigned to all event times occurring after that horizon. As the horizon advances, nonnegative event-time probability is removed from this remaining mass, so the resulting survival curve is necessarily non-increasing. PMF-based models therefore guarantee monotonicity without using the cumulative-product construction of discrete-hazard models.

Multi-task logistic regression (MTLR) models survival using a sequence of dependent binary prediction tasks defined over ordered time intervals (Yu et al., 2011). The dependence between these tasks is essential: rather than allowing arbitrary combinations of alive and failed states, MTLR assigns probability only to valid event-time configurations in which an individual is alive up to one interval and has failed thereafter. These valid configurations define a normalized distribution over event-time intervals, from which the survival curve is obtained by summing the probabilities of configurations that survive beyond each horizon. MTLR can therefore be interpreted as a structured PMF model, and its survival predictions are monotone by construction. Neural extensions of MTLR replace the linear feature representation with a neural network while retaining the same structured output distribution (Fotso, 2018).

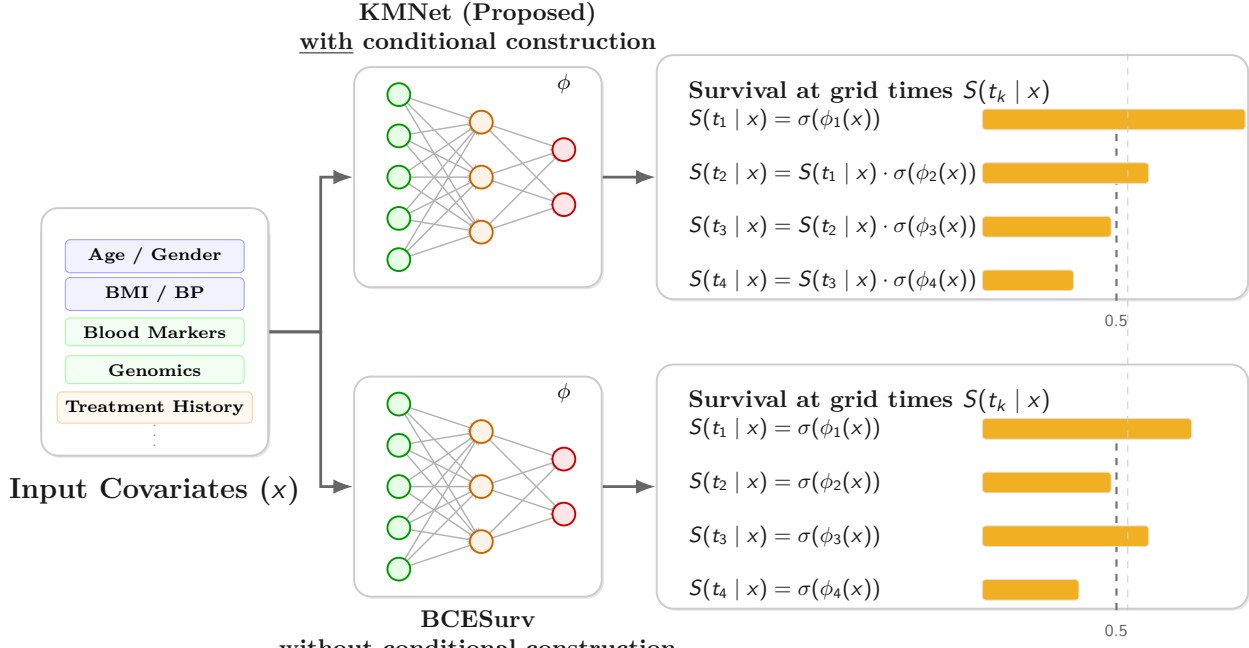

Figure 2: Conceptual comparison between KMNet and BCESurv on a discrete time grid. KMNet predicts interval-wise conditional survival probabilities $\phi_k(x)$ and constructs survival recursively as $S(t_k \mid x) = S(t_{k-1} \mid x) \cdot \sigma(\phi_k(x))$, which guarantees a non-increasing survival curve across grid times. In contrast, BCESurv predicts marginal survival values at each grid time independently as $S(t_k \mid x) = \sigma(\phi_k(x))$, which can lead to inconsistent, non-monotone survival curves.

DeepHit directly estimates the joint probability distribution of event time and event type (Lee et al., 2018). Its output layer assigns nonnegative, normalized probability mass to each cause–time combination. Cause-specific cumulative incidence is obtained by summing the corresponding probability masses up to a given horizon, while overall survival is the probability mass remaining beyond that horizon. The cumulative incidence functions are therefore non-decreasing, and the associated survival curve is non-increasing. In the absence of competing risks, DeepHit reduces to a neural PMF model.

Although monotonicity is well established in discrete-time survival modelling, different model families enforce it through different output parameterizations and training objectives. Discrete-hazard models obtain valid survival curves from products of complementary hazards, whereas MTLR and DeepHit derive monotonicity from structured probability distributions over event times. In contrast, marginal-survival BCE formulations such as BCESurv predict survival probabilities independently across time points and therefore do not guarantee non-increasing curves; among the baselines considered in this study, BCESurv is the only method without structural monotonicity. Post-hoc corrections such as cumulative-minimum and isotonic projection remove these violations, but our experiments show that they do not consistently recover the predictive performance of an end-to-end monotone formulation. Moreover, discrimination-oriented ranking objectives are available in some model families but are typically tied to cumulative event probabilities or global risk scores. These observations motivate a modular conditional-survival formulation in which monotonicity follows from the product construction, while the ranking component can be introduced and controlled independently. The proposed adopts this formulation and augments it with an event-interval conditional ranking objective.

## 3    Kaplan–Meier Net

We consider right-censored survival data $\{(X_i, Y_i, \delta_i)\}_{i=1}^N$, where $X_i \in \mathbb{R}^d$ is a covariate vector, $Y_i$ is the observed time, and $\delta_i \in \{0, 1\}$ is the event indicator. We work on a fixed, discrete time grid $0 = t_0 < t_1 < \cdots < t_J$. Intuitively, the grid partitions follow-up into intervals $(t_{j-1}, t_j]$, and the model predicts survival one interval at a time. We aim to estimate a patient-specific survival curve $S(t \mid x)$ on this grid

while respecting the non-increasing property of survival in time. Figure 2 demonstrates the conditional construction of KMNet in comparison to BCESurv.

KMNet is a neural network $\phi\colon \mathbb{R}^d \to \mathbb{R}^J$ that outputs logits $\phi(x) = (\phi_1(x), \ldots, \phi_J(x))$, where each component $\phi_j(x)$ corresponds to the interval $(t_{j-1}, t_j]$. The $j$th output is mapped through a sigmoid function to obtain a patient-specific conditional survival probability

$$p_j(x) = \sigma(\phi_j(x)) \in (0,1), \qquad j = 1, \ldots, J,$$

which we interpret as the probability of surviving the next interval given survival up to its start,

$$p_j(x) \approx \Pr(T > t_j \mid T > t_{j-1}, X = x).$$

The sigmoid ensures that each $p_j(x)$ is a valid probability in $(0, 1)$.

Given $\{p_j(x)\}_{j=1}^J$, KMNet constructs the survival curve by a Kaplan–Meier style product on the grid,

$$\widehat{S}(t_0 \mid x) = 1, \qquad \widehat{S}(t_j \mid x) = \prod_{k=1}^{j} p_k(x), \quad j = 1, \ldots, J.$$

This construction guarantees a non-increasing survival curve whenever $p_k(x) \in [0, 1]$, which holds by design under the sigmoid output.

~~To define supervision targets on the grid, each individual is assigned an index $\kappa_i \in \{1, \ldots, J\}$ indicating the first grid time greater than or equal to the observed time $Y_i$, $\kappa_i = \min\{j \in \{1, \ldots, J\} \mid Y_i \le t_j\}$. Thus, $\kappa_i$ is the discrete-time bin index of subject $i$.~~ To define the discrete-time targets, observed event times are mapped to the next grid point, whereas censoring times are mapped to the largest grid point not exceeding the censoring time, following the discrete-time label transformation used in our implementation. We denote the resulting grid index for subject $i$ by $\kappa_i$. Consequently, a subject censored at the same discretized index as an event anchor is known to have remained under observation through the anchor grid point and is therefore a valid comparator. We then define a binary survival indicator at each grid point,

$$s_{ij} = 1\{j < \kappa_i\}, \qquad i = 1, \ldots, N, \ \ j = 1, \ldots, J,$$

so that $s_{ij} = 1$ indicates subject $i$ is known to have survived beyond $t_j$, while $s_{ij} = 0$ indicates that $t_j$ lies at or after the subject's observed bin.

Learning must account for the fact that individuals only contribute to the loss while they are at risk and under observation. We introduce an at-risk-at-start indicator

$$r_{ij} = \begin{cases} 1, & j = 1, \\ s_{i,j-1}, & j = 2, \ldots, J, \end{cases}$$

so that $r_{ij} = 1$ means subject $i$ is at risk at the start of interval $(t_{j-1}, t_j]$. We further apply an exclusive censoring rule through a counting mask

$$c_{ij} = \delta_i + (1 - \delta_i)\, s_{ij}.$$

For event observations ($\delta_i = 1$), $c_{ij} = 1$ for all $j$, so the event bin $j = \kappa_i$ is eligible to contribute to the loss (with label $s_{i\kappa_i} = 0$). For censored observations ($\delta_i = 0$), $c_{ij} = s_{ij}$, so all bins $j \ge \kappa_i$ are excluded and the model is not forced to make assertions beyond the censoring bin. The final weight is

$$w_{ij} = r_{ij}\, c_{ij}.$$

Hence, $w_{ij} = 1$ exactly for subject–interval pairs that are both observable and at risk, and $w_{ij} = 0$ otherwise. Note that $r_{ij}$ enforces the sequential at-risk constraint: since $r_{ij} = s_{i,j-1}$ for $j \ge 2$, we have $w_{ij} = 0$ for all $j > \kappa_i$, so no loss is accumulated after the event or censoring bin.

The base loss is a weighted binary cross-entropy applied to conditional survival probabilities. With $p_{ij} = p_j(X_i) = \sigma(\phi_j(X_i))$, we define

$$\mathcal{L}_{\text{BCE}} = -\frac{1}{N} \sum_{i=1}^{N} \sum_{j=1}^{J} w_{ij} \Big[ s_{ij} \log(p_{ij}) + (1 - s_{ij}) \log(1 - p_{ij}) \Big].$$

This objective trains the network to estimate per-interval conditional survival while restricting supervision to time points at which an individual remains under observation and therefore at risk, with censoring handled through the weights $w_{ij}$. Intuitively, each training sample contributes gradients only for the consecutive intervals that it is known to have survived. Thus, at every discrete time point, the loss is effectively computed on the subset of individuals who are still at risk, and no supervision is imposed beyond an individual's event or censoring time. This construction is closely related to Nnet, also implemented as LogisticHazard, but differs in how the network output and training loss are expressed (Gensheimer & Narasimhan, 2019; Kvamme & Borgan, 2021). Nnet predicts interval-specific event hazards and optimizes the corresponding discrete-time negative log-likelihood. The predicted hazards are then converted into patient-specific conditional survival probabilities, which are multiplied across intervals to obtain the patient-specific survival curve. In contrast, KMNet directly predicts the conditional probability of surviving each interval and trains these outputs using a censoring-aware masked binary cross-entropy loss. KMNet is distinguished by expressing the model directly in conditional-survival form and by augmenting the base objective with the event-interval conditional ranking loss described next.

To improve risk discrimination, we augment the objective with a smooth ranking term computed at the event interval of each uncensored individual. For each event anchor $i$ with $\delta_i = 1$, define the set of comparable indices

$$\mathcal{C}_i = \Big\{ k \in \{1, \ldots, N\} \;\Big|\; \kappa_i < \kappa_k \Big\} \cup \Big\{ k \in \{1, \ldots, N\} \;\Big|\; \kappa_i = \kappa_k, \; \delta_k = 0 \Big\}.$$

The first set contains individuals whose observed bin occurs strictly after the event bin of $i$, and the second set contains individuals censored in the same bin as $i$ (who are known to have survived up to that bin).

For each event anchor $i$ with $\delta_i = 1$, we compare conditional survival probabilities at the anchor index $\kappa_i$ via

$$m_{ik} = p_{\kappa_i}(X_k) - p_{\kappa_i}(X_i), \qquad k \in \mathcal{C}_i.$$

The ranking loss uses an exponential penalty with temperature $\tau > 0$,

$$\mathcal{L}_{\text{Rank}} = \frac{1}{N} \sum_{i=1}^{N} \mathbb{1}\{\delta_i = 1\} \, \mathbb{1}\{|\mathcal{C}_i| > 0\} \, \frac{1}{|\mathcal{C}_i|} \sum_{k \in \mathcal{C}_i} \exp\Big(-\frac{m_{ik}}{\tau}\Big).$$

Minimizing $\mathcal{L}_{\text{Rank}}$ encourages $p_{\kappa_i}(X_k)$ to be larger than $p_{\kappa_i}(X_i)$ whenever individual $k$ survives longer than individual $i$, aligning the ordering of interval-conditional survival with observed event times.

In addition to treating $\tau$ as a tunable hyperparameter, we allow an automatic setting that adapts $\tau$ to the current scale of the network outputs. For a minibatch of size $B$, we collect the anchor logits $\{\phi_{\kappa_i}(X_i)\}_{i=1}^{B}$ (where $\kappa_i$ is the observed-bin index used in the loss) and set

$$\tau = \max\big(10^{-3}, \, 0.25 \cdot \text{Std}\big(\{\phi_{\kappa_i}(X_i)\}_{i=1}^{B}\big)\big),$$

computed without gradient tracking. This heuristic scales ranking margins by the empirical variability of the anchor logits, which stabilizes optimization and reduces sensitivity of the ranking term to the absolute logit scale induced by the network architecture and learning dynamics.

This ranking term differs from the ranking losses commonly used in DeepHit and related survival models, which typically rank patients using a global risk score or a cumulative quantity derived from the full survival distribution. In contrast, our ranking is local to the event interval of the anchor subject. For an event observation $i$, we compare subjects only through the conditional survival probability at $\kappa_i$, namely $p_{\kappa_i}(x)$, which corresponds to survival on the interval $(t_{\kappa_i-1}, t_{\kappa_i}]$ given survival up to $t_{\kappa_i-1}$. If two subjects have both survived up to $t_{\kappa_i-1}$, then their ordering at time $\kappa_i$ should be determined by how likely they are to

survive the next interval, rather than by a global score that aggregates information from other time regions. As a result, the ranking term directly encourages separation in the interval-specific conditional probabilities that define the Kaplan–Meier product, while preserving the interpretation of KMNet outputs as conditional survival probabilities. A mathematically equivalent DeepHit-style ranking objective, expressed using the KMNet margin notation, is provided in Appendix A.

The final training objective combines the base loss and the ranking loss using a convex weight and an additional scale factor:

$$\mathcal{L} = \alpha\,\mathcal{L}_{\text{BCE}} + (1 - \alpha)\,\lambda\,\mathcal{L}_{\text{Rank}}, \qquad \alpha \in [0, 1],\ \lambda > 0,$$

where $\alpha$ controls the trade-off between fitting the conditional survival probabilities and improving discrimination, $\lambda$ scales the ranking term, and $\tau$ is the temperature parameter inside the ranking loss. The additional factor $\lambda$ is included because the BCE and ranking components can have substantially different numerical and gradient scales: the former aggregates subject–interval contributions, whereas the latter depends on the number of comparable pairs and the exponential penalty. Thus, $\alpha$ specifies the intended balance between the two learning objectives, while $\lambda$ compensates for their scale mismatch and prevents either component from dominating optimization solely because of its magnitude. We treat $\alpha$, $\tau$, and $\lambda$ as hyperparameters and select them on the validation split. At inference time, KMNet outputs interval-wise conditional survival probabilities $\{p_j(x)\}_{j=1}^{J}$ and constructs the predicted survival curve by the cumulative product

$$\widehat{S}(t_0 \mid x) = 1, \qquad \widehat{S}(t_j \mid x) = \prod_{k=1}^{j} p_k(x), \quad j = 1, \ldots, J.$$

**Proposition 1.** *Let $\{p_j(x)\}_{j=1}^{J}$ denote the conditional survival probabilities produced by KMNet on the grid $0 = t_0 < t_1 < \cdots < t_J$ for a covariate vector $x$, with $p_j(x) \in [0, 1]$ for all $j$. Define $\widehat{S}(t_0 \mid x) = 1$ and*

$$\widehat{S}(t_j \mid x) = \prod_{k=1}^{j} p_k(x), \qquad j = 1, \ldots, J.$$

*Then the sequence $\{\widehat{S}(t_j \mid x)\}_{j=0}^{J}$ is non-increasing in $j$.*

*Proof.* For any $j \in \{0, \ldots, J - 1\}$,

$$\widehat{S}(t_{j+1} \mid x) = \widehat{S}(t_j \mid x)\,p_{j+1}(x).$$

Since $0 \leq p_{j+1}(x) \leq 1$, it follows that $\widehat{S}(t_{j+1} \mid x) \leq \widehat{S}(t_j \mid x)$. Therefore $\widehat{S}(t_j \mid x)$ is non-increasing in $j$. □

Unlike BCE-based models that directly learn marginal survival probabilities at each grid point without enforcing cross-time constraints, KMNet constructs survival curves through a product of interval-wise conditional probabilities, which guarantees valid non-increasing survival predictions by design.

## 4 Experiments

In this section, we evaluate KMNet against established neural survival baselines on multiple benchmark datasets. We first describe the datasets and preprocessing, then present the experimental protocol and evaluation metrics, and finally report comparative results along with a sensitivity analysis of key hyperparameters. ~~An ablation study is provided in Appendix ??~~

### 4.1 Datasets

We evaluate KMNet on eight widely used right-censored survival benchmarks covering diverse application domains. Support Knaus et al. (1995) is a large clinical cohort of seriously ill hospitalized patients with mortality as the endpoint. Metabric Curtis et al. (2012) and Rotterdam-Gbsg Foekens et al. (2000) are oncology benchmarks that model time-to-event outcomes in breast cancer cohorts. Flchain Dispenzieri et al.

(2012) is a clinical cohort related to serum free light chain measurements and models time to death. Nwtco Breslow & Chatterjee (1999) is a pediatric oncology benchmark from the National Wilms Tumor Study and considers time-to-relapse outcomes. We additionally include two datasets from the Northern Alberta Cancer Database: Nacd contains patients across multiple cancer sites, while Nacd-Col is the colorectal cancer subset of the same registry; the two share the same feature definition while differing in disease focus and cohort composition Haider et al. (2020). Finally, Recidivism Rossi et al. (2013) is a non-clinical dataset that models time to re-incarceration after release. Detailed dataset characteristics and summary statistics are reported in Table 1.

| Dataset | Size | Feature | Censoring | Median/IQR/Range | |
| Name | (N) | (d) | (%) | Event Time | Censored Time |
|---|---|---|---|---|---|
| Flchain | 6524 | 8 | 70 | 4621/734/5165 | 2084/2343/4998 |
| Metabric | 1904 | 9 | 42 | 158/109/337 | 86/102/355 |
| Nacd | 2402 | 48 | 36 | 33/35/84 | 12/17/81 |
| Nacd-Col | 950 | 48 | 52 | 33/34/83 | 17/21/81 |
| Nwtco | 4028 | 6 | 86 | 2323/2580/6204 | 280/337/4162 |
| Recidivism | 1445 | 14 | 61 | 19/64/76 | 74/6/11 |
| Rotterdam-Gbsg | 2232 | 7 | 43 | 75/32/87 | 24/26/82 |
| Support | 8873 | 14 | 32 | 918/917/1685 | 57/236/1941 |

Table 1: Details of the datasets used in the experiments. The reported number of features is after preprocessing.

## 4.2 Experimental setup

For each dataset, we created a random split in which 80% of the samples were used for model development and the remaining 20% were held out as an independent test set. From the development split, we further set aside 10% of the samples as a validation set used only for hyperparameter optimization and model selection. Model performance was evaluated using the time-dependent concordance (Concordance) (Heagerty et al., 2000; Antolini et al., 2005), integrated Brier score (IBS) (Graf et al., 1999; Gerds & Schumacher, 2006; Kvamme & Borgan, 2019) and D-Calibration (Haider et al., 2020), with definitions provided in Appendix B. Hyperparameters were tuned with 200 trials via Bayesian optimization Bergstra et al. (2011), and the full search spaces are reported in Appendix C. After selecting the best hyperparameters, we refit the model using the training portion of the split and report results on the held-out test set. This procedure was repeated across multiple five random splits, and we report the mean and standard deviation across runs. For Rotterdam-Gbsg, we follow the standard protocol in which the Rotterdam cohort is used for training and the GBSG cohort is used for testing, and results are reported for this single split.

We compare KMNet with seven established neural survival baselines. BCESurv Kvamme & Borgan (2019) is a discrete-time MLP that learns marginal survival probabilities on a fixed grid using a censoring-aware binary cross-entropy objective. CoxCC Kvamme et al. (2019) is a Cox proportional hazards network trained with a case-control approximation to the partial likelihood, and CoxTime Kvamme et al. (2019) extends this approach by allowing time-dependent effects. DeepHit Lee et al. (2018) is a discrete-time model that learns the event-time distribution and includes a ranking component for discrimination. DeepSurv Katzman et al. (2018) is a neural Cox model that replaces the linear predictor with a deep network while optimizing the Cox partial likelihood. MTLR Yu et al. (2011) models survival through a sequence of logistic regressions across time and outputs a full discrete-time survival distribution. NNet Gensheimer & Narasimhan (2019) models the hazard as piecewise constant over time intervals and is trained through likelihood-based objectives.

## 4.3 Results

Tables 2 and 3 report the mean and standard deviation of the time-dependent concordance and the integrated Brier score (IBS) across repeated random splits. For each dataset, higher concordance indicates better discrimination, while lower IBS indicates better overall accuracy of the predicted survival probabilities. We also report the average rank across datasets to summarize overall performance.

**Discrimination** KMNet demonstrate ~~achieves the best overall~~ competitive performance with best average rank (1.62) for discrimination as shown in Table 2. It attains the top concordance on Nacd, Nwtco, and Support, and remains competitive on the remaining datasets where the best model varies between DeepSurv, CoxTime, and DeepHit. Compared with BCESurv, KMNet achieves higher concordance on every dataset, with ~~strong gains~~larger observed improvements on Nacd-Col, Metabric, Recidivism, and Support. These results indicate that learning interval-wise conditional survival probabilities together with the proposed conditional ranking term yields consistent improvements in risk stratification across heterogeneous censoring levels and application domains.

**Calibration and overall accuracy** Table 3 shows that KMNet obtains competerive performance with the best average rank (2.38) in terms of IBS. The strongest competing baseline for IBS is NNet, which achieves the lowest score on several datasets, but KMNet remains consistently close and attains the best IBS on Rotterdam-Gbsg. Relative to BCESurv, KMNet reduces IBS across all datasets, with the largest reduction observed on Nwtco, and smaller but consistent reductions on Flchain, Metabric, Nacd, and Nacd-Col. On Support, the differences in IBS between methods are small, suggesting that the main gains on this dataset arise from improved discrimination rather than large changes in average probability error.

| Dataset | BCESurv | CoxCC | CoxTime | DeepHit | DeepSurv | NNet | MTLR | KMNet |
|---|---|---|---|---|---|---|---|---|
| Flchain | $0.782_{\pm 0.009}$ | $0.793_{\pm 0.009}$ | $0.793_{\pm 0.009}$ | $0.788_{\pm 0.009}$ | $\mathbf{0.795}_{\pm \mathbf{0.007}}$ | $0.789_{\pm 0.015}$ | $0.783_{\pm 0.013}$ | $\underline{0.793}_{\pm 0.006}$ |
| Metabric | $0.638_{\pm 0.022}$ | $0.650_{\pm 0.015}$ | $\mathbf{0.665}_{\pm \mathbf{0.012}}$ | $0.660_{\pm 0.027}$ | $0.643_{\pm 0.012}$ | $0.653_{\pm 0.027}$ | $0.661_{\pm 0.029}$ | $\underline{0.664}_{\pm 0.007}$ |
| Nacd | $0.745_{\pm 0.018}$ | $0.755_{\pm 0.021}$ | $0.756_{\pm 0.017}$ | $0.753_{\pm 0.012}$ | $0.755_{\pm 0.015}$ | $0.750_{\pm 0.014}$ | $\underline{0.759}_{\pm 0.008}$ | $\mathbf{0.760}_{\pm \mathbf{0.015}}$ |
| Nacd-Col | $0.673_{\pm 0.035}$ | $0.703_{\pm 0.053}$ | $0.692_{\pm 0.016}$ | $\mathbf{0.722}_{\pm \mathbf{0.031}}$ | $0.697_{\pm 0.040}$ | $0.680_{\pm 0.056}$ | $0.698_{\pm 0.036}$ | $\underline{0.704}_{\pm 0.016}$ |
| Nwtco | $0.695_{\pm 0.043}$ | $0.710_{\pm 0.023}$ | $0.709_{\pm 0.025}$ | $\underline{0.712}_{\pm 0.018}$ | $0.709_{\pm 0.021}$ | $0.704_{\pm 0.036}$ | $0.703_{\pm 0.023}$ | $\mathbf{0.713}_{\pm \mathbf{0.024}}$ |
| Recidivism | $0.615_{\pm 0.031}$ | $0.614_{\pm 0.029}$ | $0.623_{\pm 0.044}$ | $0.599_{\pm 0.024}$ | $\mathbf{0.640}_{\pm \mathbf{0.021}}$ | $0.603_{\pm 0.025}$ | $0.611_{\pm 0.026}$ | $\underline{0.637}_{\pm 0.012}$ |
| Rotterdam-Gbsg | $0.677_{\pm 0.000}$ | $0.660_{\pm 0.000}$ | $\mathbf{0.693}_{\pm \mathbf{0.000}}$ | $0.680_{\pm 0.000}$ | $0.675_{\pm 0.000}$ | $0.652_{\pm 0.000}$ | $0.660_{\pm 0.000}$ | $\underline{0.682}_{\pm 0.000}$ |
| Support | $0.619_{\pm 0.013}$ | $0.605_{\pm 0.007}$ | $0.610_{\pm 0.008}$ | $\underline{0.636}_{\pm 0.010}$ | $0.609_{\pm 0.009}$ | $0.624_{\pm 0.018}$ | $0.623_{\pm 0.016}$ | $\mathbf{0.637}_{\pm \mathbf{0.006}}$ |
| Average Rank | 6.62 | 5.00 | $\underline{3.50}$ | 4.00 | 4.25 | 6.00 | 5.00 | **1.62** |

Table 2: Time-dependent concordance performance across eight benchmark survival datasets. Results are reported as mean $\pm$ standard deviation across repeated splits; higher values indicate better discrimination. The best result for each dataset is shown in bold, the second-best is underlined, and the final row reports the average rank across datasets, where lower is better.

| Dataset | BCESurv | CoxCC | CoxTime | DeepHit | DeepSurv | NNet | MTLR | KMNet |
|---|---|---|---|---|---|---|---|---|
| Flchain | $0.105_{\pm 0.003}$ | $0.101_{\pm 0.002}$ | $0.100_{\pm 0.003}$ | $0.101_{\pm 0.003}$ | $0.101_{\pm 0.003}$ | $\mathbf{0.098}_{\pm \mathbf{0.003}}$ | $0.100_{\pm 0.002}$ | $\underline{0.099}_{\pm 0.002}$ |
| Metabric | $0.176_{\pm 0.017}$ | $0.163_{\pm 0.007}$ | $\underline{0.160}_{\pm 0.003}$ | $0.161_{\pm 0.005}$ | $0.166_{\pm 0.015}$ | $\mathbf{0.157}_{\pm \mathbf{0.001}}$ | $0.163_{\pm 0.004}$ | $0.162_{\pm 0.004}$ |
| Nacd | $0.155_{\pm 0.006}$ | $0.144_{\pm 0.006}$ | $0.144_{\pm 0.008}$ | $0.145_{\pm 0.006}$ | $\mathbf{0.139}_{\pm \mathbf{0.004}}$ | $0.147_{\pm 0.003}$ | $0.149_{\pm 0.007}$ | $\underline{0.142}_{\pm 0.004}$ |
| Nacd-Col | $0.187_{\pm 0.030}$ | $0.188_{\pm 0.013}$ | $0.175_{\pm 0.013}$ | $0.173_{\pm 0.013}$ | $0.180_{\pm 0.009}$ | $\mathbf{0.171}_{\pm \mathbf{0.012}}$ | $0.178_{\pm 0.024}$ | $\underline{0.173}_{\pm 0.005}$ |
| Nwtco | $0.126_{\pm 0.008}$ | $0.103_{\pm 0.007}$ | $0.103_{\pm 0.005}$ | $0.102_{\pm 0.015}$ | $0.103_{\pm 0.005}$ | $\mathbf{0.083}_{\pm \mathbf{0.005}}$ | $0.090_{\pm 0.005}$ | $\underline{0.086}_{\pm 0.005}$ |
| Recidivism | $0.183_{\pm 0.007}$ | $0.177_{\pm 0.009}$ | $\mathbf{0.175}_{\pm \mathbf{0.004}}$ | $0.178_{\pm 0.008}$ | $0.179_{\pm 0.014}$ | $0.179_{\pm 0.006}$ | $0.183_{\pm 0.008}$ | $\underline{0.177}_{\pm 0.005}$ |
| Rotterdam-Gbsg | $0.179_{\pm 0.000}$ | $0.174_{\pm 0.000}$ | $0.170_{\pm 0.000}$ | $0.175_{\pm 0.000}$ | $0.167_{\pm 0.000}$ | $0.169_{\pm 0.000}$ | $\underline{0.167}_{\pm 0.000}$ | $\mathbf{0.165}_{\pm \mathbf{0.000}}$ |
| Support | $0.192_{\pm 0.001}$ | $\underline{0.191}_{\pm 0.001}$ | $0.192_{\pm 0.003}$ | $0.191_{\pm 0.002}$ | $0.192_{\pm 0.003}$ | $\mathbf{0.190}_{\pm \mathbf{0.001}}$ | $0.192_{\pm 0.001}$ | $0.191_{\pm 0.001}$ |
| Mean Rank | 7.75 | 4.88 | 4.12 | 4.50 | 5.12 | $\underline{2.50}$ | 4.75 | **2.38** |

Table 3: Integrated Brier score performance across eight benchmark survival datasets. Results are reported as mean $\pm$ standard deviation across repeated splits; lower values indicate better calibration and prediction accuracy. The best result for each dataset is shown in bold, the second-best is underlined, and the final row reports the average rank across datasets, where lower is better.

**D-calibration** Table 4 complements the IBS analysis by evaluating whether the predicted survival distributions satisfy D-calibration. KMNet yields (p>0.05) on seven of the eight datasets, indicating that D-calibration is not rejected at the (5%) level except on Support. This represents a substantial improvement over BCESurv, for which the null hypothesis is rejected on seven datasets and retained only on Rotterdam-

| Dataset | BCESurv | CoxCC | CoxTime | DeepHit | DeepSurv | NNet | MTLR | KMNet |
|---|---|---|---|---|---|---|---|---|
| Flchain | $<0.001$ | **0.998** | **0.772** | **0.894** | **0.975** | **0.988** | **0.754** | **0.776** |
| Metabric | $<0.001$ | **0.276** | **0.859** | **0.211** | **0.527** | 0.004 | 0.036 | **0.179** |
| Nacd | $<0.001$ | **0.498** | 0.021 | **0.513** | **0.508** | 0.003 | **0.281** | **0.133** |
| Nacd-Col | $<0.001$ | **0.479** | **0.904** | **0.138** | **0.075** | **0.122** | 0.001 | **0.054** |
| Nwtco | $<0.001$ | **0.997** | **1.000** | **0.880** | **1.000** | **1.000** | **1.000** | **0.998** |
| Recidivism | $<0.001$ | **0.954** | **0.488** | $<0.001$ | **0.767** | **0.315** | **0.864** | **0.080** |
| Rotterdam-Gbsg | **0.504** | **0.609** | **0.450** | 0.008 | **0.488** | **0.058** | **0.600** | **0.789** |
| Support | $<0.001$ | **0.349** | 0.022 | $<0.001$ | **0.104** | $<0.001$ | $<0.001$ | $<0.001$ |

Table 4: D-calibration $p$-values across datasets and models. Values greater than 0.05 (**bold**) indicate that the null hypothesis of D-calibration is not rejected at the 5% significance level; they should not be interpreted as a ranked measure of calibration quality.

Gbsg. KMNet also shows broadly competitive distributional calibration relative to the other neural survival models, although CoxCC and DeepSurv satisfy the test on all eight datasets. These results suggest that the conditional-survival parameterization of KMNet generally produces survival distributions that are compatible with the observed event-time frequencies across probability intervals. Nevertheless, a large D-calibration ($p$)-value should not be interpreted as evidence that one model is better calibrated than another; it only indicates insufficient evidence to reject the D-calibration hypothesis. The D-calibration results should therefore be considered together with the horizon-specific reliability curves, calibration slopes, and IBS values. The uniformity histograms for every dataset are deferred to Appendix D (see Figure 9, 10, 11, 12, 13, 14, 15, 16)

Overall, KMNet achieves the best average rank for concordance and integrated Brier score performance, although NNet attains lower IBS on several individual datasets. The D-calibration results further show that the null hypothesis of distributional calibration is not rejected for KMNet on seven of the eight datasets. Taken together, these findings suggest that KMNet is a useful approach for predicting patient-specific survival curves while guaranteeing that the estimated survival probabilities are non-increasing by construction. Additional diagnostic results are provided in Appendix D, including horizon-specific calibration intercept and slope in Table 15 and 16 respectively, L1 loss in Table 17, reliability curves in Figure 17, IPCW Brier scores over time in Figure 18, time-dependent AUC curves in Figure 19.

| Dataset | Upward steps $\Delta_t \widehat{S} > 0.01$ (%) | Any upward step (%) | Ambiguous median (%) |
|---|---|---|---|
| Flchain | 55.4 | 100.0 | 47.4 |
| Metabric | 77.7 | 100.0 | 54.1 |
| Nacd | 6.0 | 100.0 | 6.7 |
| Nacd-Col | 99.5 | 100.0 | 80.0 |
| Nwtco | 0.9 | 100.0 | 3.6 |
| Recidivism | 38.8 | 100.0 | 40.1 |
| Rotterdam-Gbsg | 0.4 | 94.5 | 3.2 |
| Support | 0.3 | 100.0 | 7.7 |

Table 5: Non-monotonicity of BCESurv predictions across datasets. Reported are the percentage of adjacent time intervals with an upward survival-probability change greater than 0.01, the percentage of individuals with at least one upward step, and the percentage with an ambiguous predicted median survival time due to multiple crossings of the 0.5 threshold.

BCESurv is the most direct comparator to KMNet because it predicts marginal survival probabilities separately at the discretized time points and produces non-monotone survival curves, as shown in Table 5. To determine whether this limitation can be addressed without modifying or retraining the model, we apply two post-hoc monotonicity corrections to the predictions of the trained BCESurv model. The cumulative-minimum correction (+cummin) replaces each prediction by its running minimum, ($\widetilde{S}(t \mid \mathbf{x}) = \min_{t' \leq t} \widehat{S}(t' \mid \mathbf{x})$), thereby producing the smallest pointwise downward adjustment required to obtain a non-increasing curve. The isotonic-regression correction (+isotonic) instead projects each predicted sequence onto the set

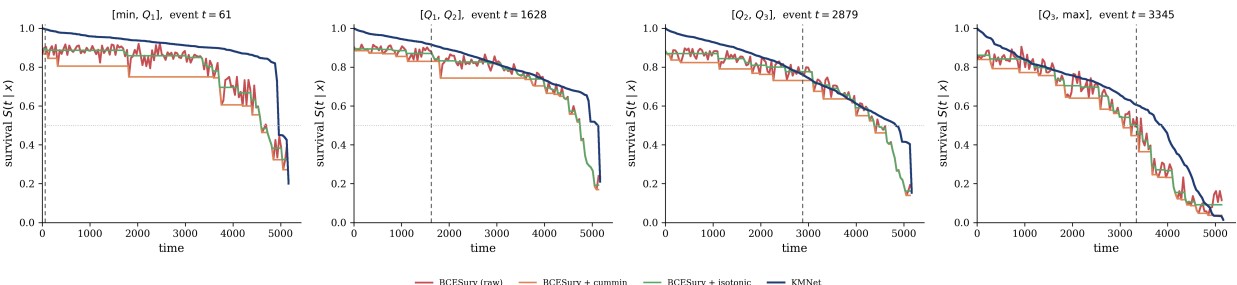

Figure 3: Predicted survival curves for four Flchain patients, one drawn at random from each event-time quartile band $[\min, Q_1]$, $[Q_1, Q_2]$, $[Q_2, Q_3]$, $[Q_3, \max]$; all four are uncensored (their observed event time is the dashed vertical line, and the dotted horizontal line marks $S = 0.5$). Each panel shows raw BCESurv (red), its two post-hoc monotonizations BCESurv+`cummin` (orange) and BCESurv+`isotonic` (green), and KMNet (dark blue). BCESurv's marginal per-interval estimates are non-monotone—the curves rise and fall, so the survival function is not a valid (non-increasing) curve and its median survival time is ambiguous. The post-processed variants restore monotonicity but merely trace BCESurv's mis-shaped curve, whereas KMNet is monotone by construction.

of non-increasing sequences under a least-squares criterion using the pool-adjacent-violators algorithm Ayer et al. (1955). Figure 3 illustrates the original BCESurv prediction, the two corrected curves, and the corresponding KMNet prediction. Both corrections are applied to the same trained BCESurv outputs, thereby guaranteeing monotonicity without changing the underlying model.

Table 6 shows that enforcing monotonicity after prediction does not consistently improve the predictive performance of BCESurv. Isotonic regression generally preserves concordance better than the cumulative-minimum correction and improves upon the original BCESurv concordance on several datasets. However, both post-processing methods produce higher IBS than the original BCESurv model on most datasets. For example, cumulative-minimum correction increases the IBS on Nacd-Col from (0.187) to (0.212). The null hypothesis of D-calibration is rejected for the original BCESurv model on seven of the eight datasets and remains rejected on the same seven datasets after applying either correction, with only minor changes in the corresponding (p)-values (see Table 14). These results show that post-processing successfully removes monotonicity violations but does not consistently improve D-calibration or IBS. In contrast, KMNet achieves higher concordance and lower IBS than the original and post-processed BCESurv variants across all eight datasets, suggesting that incorporating monotonicity directly into the model formulation is more effective than correcting independently estimated marginal survival probabilities after training. Nevertheless, post-hoc correction remains a simple alternative when retraining the model is not feasible.

# 5    Ablation Study

We conduct two complementary ablation analyses to clarify which components of KMNet are responsible for its empirical behaviour. First, we isolate the proposed event-interval conditional ranking objective by comparing the complete KMNet formulation with variants that remove the ranking term, replace it with a global cumulative-distribution ranking objective, or fix its weighting coefficient. Second, while retaining the conditional-ranking formulation, we vary the base loss, ranking space, and pairwise penalty to assess the sensitivity of KMNet to individual objective-design choices. All variants preserve the same network architecture and cumulative-product survival construction, so the comparisons focus specifically on the training objective.

## 5.1    Ranking Ablation for KMNet

To isolate the role and formulation of the ranking component, we compare the complete KMNet objective with three controlled alternatives. The no-ranking variant sets the ranking coefficient to ($\lambda = 0$), thereby retaining only the censoring-aware base loss. The global-CDF variant replaces the proposed event-interval conditional comparison with a DeepHit-style comparison based on the cumulative event probability evaluated at the

| Dataset | BCESurv family | | | KMNet |
|---|---|---|---|---|
| | BCESurv | +`cummin` | +`isotonic` | |
| *Concordance* ↑ | | | | |
| Flchain | $0.782 \pm 0.009$ | $0.782 \pm 0.007$ | $0.782 \pm 0.007$ | $\mathbf{0.793} \pm \mathbf{0.006}$ |
| Metabric | $0.638 \pm 0.022$ | $0.643 \pm 0.030$ | $\underline{0.645} \pm 0.030$ | $\mathbf{0.664} \pm \mathbf{0.007}$ |
| Nacd | $0.745 \pm 0.018$ | $0.746 \pm 0.025$ | $\underline{0.748} \pm 0.020$ | $\mathbf{0.760} \pm \mathbf{0.015}$ |
| Nacd-Col | $0.673 \pm 0.035$ | $0.657 \pm 0.042$ | $\underline{0.698} \pm 0.024$ | $\mathbf{0.704} \pm \mathbf{0.016}$ |
| Nwtco | $0.695 \pm 0.043$ | $0.692 \pm 0.051$ | $\underline{0.705} \pm 0.035$ | $\mathbf{0.713} \pm \mathbf{0.024}$ |
| Recidivism | $\underline{0.615} \pm 0.031$ | $0.603 \pm 0.036$ | $0.606 \pm 0.037$ | $\mathbf{0.637} \pm \mathbf{0.012}$ |
| Rotterdam-Gbsg | $\underline{0.677} \pm 0.000$ | $0.667 \pm 0.000$ | $0.666 \pm 0.000$ | $\mathbf{0.682} \pm \mathbf{0.000}$ |
| Support | $0.619 \pm 0.013$ | $0.619 \pm 0.004$ | $\underline{0.620} \pm 0.004$ | $\mathbf{0.637} \pm \mathbf{0.006}$ |
| Mean rank | 3.06 | 3.44 | 2.50 | **1.00** |
| *Integrated Brier score* ↓ | | | | |
| Flchain | $0.105 \pm 0.003$ | $0.114 \pm 0.004$ | $0.114 \pm 0.004$ | $\mathbf{0.099} \pm \mathbf{0.002}$ |
| Metabric | $\underline{0.176} \pm 0.017$ | $0.189 \pm 0.008$ | $0.188 \pm 0.008$ | $\mathbf{0.162} \pm \mathbf{0.004}$ |
| Nacd | $\underline{0.155} \pm 0.006$ | $0.161 \pm 0.010$ | $0.161 \pm 0.009$ | $\mathbf{0.142} \pm \mathbf{0.004}$ |
| Nacd-Col | $\underline{0.187} \pm 0.030$ | $0.212 \pm 0.024$ | $0.190 \pm 0.011$ | $\mathbf{0.173} \pm \mathbf{0.005}$ |
| Nwtco | $\underline{0.126} \pm 0.008$ | $0.191 \pm 0.052$ | $0.167 \pm 0.009$ | $\mathbf{0.086} \pm \mathbf{0.005}$ |
| Recidivism | $\underline{0.183} \pm 0.007$ | $0.193 \pm 0.012$ | $0.189 \pm 0.011$ | $\mathbf{0.177} \pm \mathbf{0.005}$ |
| Rotterdam-Gbsg | $0.179 \pm 0.000$ | $\underline{0.176} \pm 0.000$ | $\underline{0.176} \pm 0.000$ | $\mathbf{0.165} \pm \mathbf{0.000}$ |
| Support | $\underline{0.192} \pm 0.001$ | $0.207 \pm 0.005$ | $0.207 \pm 0.005$ | $\mathbf{0.191} \pm \mathbf{0.001}$ |
| Mean rank | 2.25 | 3.63 | 3.13 | **1.00** |

Table 6: Comparison of KMNet with the original BCESurv model and two post-hoc monotonicity corrections applied to BCESurv predictions. Results are reported as mean ± standard deviation across repeated splits. `cummin` replaces each predicted survival curve by its cumulative minimum, whereas isotonic applies `isotonic` regression along the time grid. The best result overall is shown in **bold**, and the best result within the BCESurv family is underlined. Higher concordance and lower integrated Brier score (IBS) are better; lower mean rank is better.

anchor individual's event time. The fixed-weight variant retains the proposed conditional ranking quantity but sets its coefficient to ($\lambda = 1$), instead of selecting it through the tuning procedure used for the complete model. All variants are evaluated using the same data splits, random seeds, backbone, discretization grid, tuning budget, and evaluation procedure. Tables 7 and 8 report concordance and integrated Brier score, respectively, with KMNet included as the reference configuration.

| Dataset | KMNet | No ranking ($\lambda = 0$) | Global-CDF ranking (DeepHit-style) | Fixed Rank Scale ($\lambda = 1$) |
|---|---|---|---|---|
| Flchain | 0.793 | 0.783 (−1.26%) | **0.787** (−0.76%) | 0.783 (−1.26%) |
| Metabric | 0.664 | 0.649 (−2.26%) | **0.670** (+0.90%) | 0.655 (−1.36%) |
| Nacd | 0.760 | 0.746 (−1.84%) | **0.749** (−1.45%) | 0.746 (−1.84%) |
| Nacd-Col | 0.704 | 0.665 (−5.54%) | 0.707 (+0.43%) | **0.710** (+0.85%) |
| Nwtco | 0.713 | 0.695 (−2.52%) | **0.700** (−1.82%) | 0.690 (−3.23%) |
| Recidivism | 0.637 | 0.586 (−8.01%) | **0.603** (−5.34%) | 0.596 (−6.44%) |
| Rotterdam-Gbsg | 0.682 | 0.654 (−4.11%) | **0.672** (−1.47%) | 0.653 (−4.25%) |
| Support | 0.637 | 0.617 (−3.14%) | **0.629** (−1.26%) | 0.625 (−1.88%) |

Table 7: Ranking-loss ablation for KMNet in terms of time-dependent concordance. The original KMNet results from Table 2 are included as reference values. Parenthesized values report the relative percentage change with respect to the corresponding KMNet result. Improvements are shown in green and decreases in red. Higher concordance is better, and the best ablation result for each dataset is shown in **bold**.

The ranking ablation provides consistent descriptive evidence that both the presence and the formulation of the ranking term matter. Removing ranking reduces concordance on all eight datasets, with relative decreases ranging from (1.26%) on Flchain to (8.01%) on Recidivism, and also produces a higher IBS on

| Dataset | KMNet | No ranking ($\lambda = 0$) | Global-CDF ranking (DeepHit-style) | Fixed Rank Scale ($\lambda = 1$) |
|---|---|---|---|---|
| Flchain | 0.099 | **0.102** (+3.03%) | **0.102** (+3.03%) | **0.102** (+3.03%) |
| Metabric | 0.162 | **0.169** (+4.32%) | 0.171 (+5.56%) | 0.171 (+5.56%) |
| Nacd | 0.142 | 0.146 (+2.82%) | **0.145** (+2.11%) | 0.150 (+5.63%) |
| Nacd-Col | 0.173 | **0.185** (+6.94%) | 0.188 (+8.67%) | 0.191 (+10.40%) |
| Nwtco | 0.086 | **0.087** (+1.16%) | 0.088 (+2.33%) | 0.093 (+8.14%) |
| Recidivism | 0.177 | 0.185 (+4.52%) | **0.183** (+3.39%) | 0.186 (+5.08%) |
| Rotterdam-Gbsg | 0.165 | 0.175 (+6.06%) | 0.174 (+5.45%) | **0.171** (+3.64%) |
| Support | 0.191 | **0.193** (+1.05%) | 0.196 (+2.62%) | 0.194 (+1.57%) |

Table 8: Ranking-loss ablation for KMNet in terms of integrated Brier score (IBS). The original KMNet results from Table 3 are included as reference values. Parenthesized values report the relative percentage change with respect to the corresponding KMNet result. Red values indicate an increase in IBS and green values indicate a decrease. Lower IBS is better, and the best ablation result for each dataset is shown in **bold**.

every dataset. Compared with the global-CDF formulation, the proposed event-interval conditional ranking achieves higher concordance on six of the eight datasets and lower IBS on all eight, although the global-CDF variant performs slightly better on Metabric and Nacd-Col in concordance. Fixing the ranking coefficient at ($\lambda = 1$) similarly yields lower concordance on seven datasets and higher IBS on all eight, indicating that the weighting of the ranking component should be adapted to the dataset rather than held constant. Overall, these results Support the proposed conditional ranking formulation as a more consistent choice than either removing ranking or applying ranking to a global cumulative quantity. This show consistent empirical improvement.

## 5.2 Base Loss, Ranking Space and Penalty

In addition to the main configuration of KMNet, we study how individual design choices in the training objective affect discrimination and calibration. Unless stated otherwise, all ablations use the same network architecture, discretization grid, optimizer, early stopping protocol, and evaluation pipeline as in the main experiments. The primary model used in this work sets the base loss to masked BCE on the discrete-time alive indicators, and augments it with a conditional ranking objective computed on per-interval conditional survival probabilities. Concretely, the default configuration is base loss BCE, ranking mode conditional, ranking space probability, and an exponential ranking penalty. We consider the following controlled variations.

1. Conditional ranking compared to global CDF ranking. The conditional variant ranks patients using the per-interval conditional survival probability at the event interval of the anchor individual. The global variant ranks using a cumulative distribution comparison evaluated at the anchor time, which depends on the full product-form survival curve.

2. Exponential compared to softplus ranking penalties. The exponential penalty imposes stronger separation between mis-ordered pairs, while the softplus penalty provides a smoother and numerically stable alternative.

3. Ranking in probability space compared to logit space. In conditional ranking, pairwise margins can be formed either on $\sigma(\phi)$ or directly on $\phi$, which changes the scale of the ranking gradients.

4. BCE compared to NLL implementations of the base objective. In our implementation the masked BCE-with-logits formulation corresponds to the same exclusive discrete-time likelihood as the NLL (NNet uses NLL loss so it is equivalent to Nnet + Ranking) form, but we include both for completeness since they may differ in numerical behavior.

Table 9 summarizes the ablated configurations considered in this study and highlights how each variant deviates from the primary model.

| Variant | Base loss | Rank space | Penalty |
|---------|-----------|------------|---------|
| KMNET-NLS | NLL | Logit | Softplus |
| KMNET-NLE | NLL | Logit | Exp |
| KMNET-NPS | NLL | Prob | Softplus |
| KMNET-NPE | NLL | Prob | Exp |
| KMNET-BLS | BCE | Logit | Softplus |
| KMNET-BLE | BCE | Logit | Exp |
| KMNET-BPS | BCE | Prob | Softplus |
| KMNET-BPE | BCE | Prob | Exp |

Table 9: Ablations of KMNet under the fixed conditional-ranking formulation. The primary configuration is KMNET-BPE.

| Dataset | KMNet-NLS | KMNet-NLE | KMNet-NPS | KMNet-NPE | KMNet-BLS | KMNet-BLE | KMNet-BPS | KMNet-BPE |
|---------|-----------|-----------|-----------|-----------|-----------|-----------|-----------|-----------|
| Flchain | $0.791_{\pm 0.008}$ | $0.790_{\pm 0.009}$ | $0.788_{\pm 0.008}$ | $\underline{0.793}_{\pm 0.005}$ | $0.791_{\pm 0.008}$ | $0.790_{\pm 0.007}$ | $0.791_{\pm 0.006}$ | $\mathbf{0.793}_{\pm \mathbf{0.006}}$ |
| Metabric | $0.659_{\pm 0.016}$ | $0.660_{\pm 0.008}$ | $0.661_{\pm 0.017}$ | $0.658_{\pm 0.017}$ | $0.653_{\pm 0.006}$ | $0.652_{\pm 0.024}$ | $\underline{0.664}_{\pm 0.016}$ | $\mathbf{0.664}_{\pm \mathbf{0.007}}$ |
| Nacd | $0.752_{\pm 0.019}$ | $0.757_{\pm 0.011}$ | $0.756_{\pm 0.014}$ | $0.752_{\pm 0.023}$ | $\underline{0.758}_{\pm 0.015}$ | $0.757_{\pm 0.013}$ | $0.757_{\pm 0.015}$ | $\mathbf{0.760}_{\pm \mathbf{0.015}}$ |
| Nacd-Col | $0.700_{\pm 0.055}$ | $\mathbf{0.727}_{\pm \mathbf{0.048}}$ | $\underline{0.711}_{\pm 0.038}$ | $0.690_{\pm 0.025}$ | $0.711_{\pm 0.056}$ | $0.696_{\pm 0.038}$ | $0.696_{\pm 0.036}$ | $0.704_{\pm 0.016}$ |
| Nwtco | $\underline{0.714}_{\pm 0.025}$ | $0.714_{\pm 0.014}$ | $0.714_{\pm 0.019}$ | $0.706_{\pm 0.026}$ | $0.713_{\pm 0.026}$ | $0.699_{\pm 0.038}$ | $\mathbf{0.718}_{\pm \mathbf{0.014}}$ | $0.713_{\pm 0.024}$ |
| Recidivism | $\underline{0.631}_{\pm 0.022}$ | $0.619_{\pm 0.024}$ | $0.630_{\pm 0.017}$ | $0.609_{\pm 0.023}$ | $0.616_{\pm 0.023}$ | $0.621_{\pm 0.034}$ | $0.612_{\pm 0.023}$ | $\mathbf{0.637}_{\pm \mathbf{0.012}}$ |
| Rotterdam-Gbsg | $\underline{0.681}_{\pm 0.000}$ | $0.673_{\pm 0.004}$ | $0.678_{\pm 0.003}$ | $0.680_{\pm 0.000}$ | $0.669_{\pm 0.007}$ | $0.677_{\pm 0.000}$ | $0.671_{\pm 0.003}$ | $\mathbf{0.682}_{\pm \mathbf{0.001}}$ |
| Support | $0.623_{\pm 0.010}$ | $0.625_{\pm 0.017}$ | $\underline{0.630}_{\pm 0.002}$ | $0.621_{\pm 0.019}$ | $0.621_{\pm 0.014}$ | $0.623_{\pm 0.015}$ | $0.616_{\pm 0.022}$ | $\mathbf{0.637}_{\pm \mathbf{0.006}}$ |

Table 10: Concordance (Higher is Better) (mean ± std)

**Ablation analysis.** Tables 10 and 11 compare eight KMNet variants obtained by changing only the training objective while keeping the same MLP backbone, discretization scheme, and Kaplan–Meier product construction for survival prediction. Overall, the primary configuration KMNET-BPE (BCE base loss with probability-space conditional ranking and exponential penalty) achieves the best average rank for concordance and remains among the top variants for calibration, indicating that the proposed ranking formulation is robust to the choice of the base likelihood surrogate. The gains in concordance are consistent on Support, Recidivism, Metabric, and Rotterdam-Gbsg, suggesting that ranking directly in the probability space aligns well with the patient-specific conditional survival outputs that are multiplied to form $\widehat{S}(t \mid x)$. In contrast, switching to NLL (KMNET-N*) or to logit-space ranking (*L*) typically produces smaller or inconsistent improvements, which is expected because the rank loss is applied at the event interval and interacts with the sigmoid nonlinearity and the multiplicative survival construction. The few exceptions are informative: on Nacd-Col, KMNET-NLE attains the best concordance and strong IBS, while on NWTCO the best concordance is reached by KMNET-BPS but the best IBS is achieved by KMNET-BLS, indicating that datasets with different censoring patterns or time-resolution requirements can favor smoother penalties (softplus) or logit comparisons for stability. Importantly, differences in IBS across variants are generally modest (often within the reported standard deviations), whereas concordance is more sensitive to the ranking design, supporting the view that the conditional ranking term primarily affects discrimination while the base loss dominates calibration. This behavior is consistent with the architecture of KMNet: calibration is largely shaped by the per-interval supervision from the base loss, while discrimination is driven by how the rank loss separates conditional survival scores at the event interval.

## 6   Statistical significance

We assess whether the observed differences between methods are statistically significant using the Friedman test followed by a Nemenyi post hoc analysis, visualized through the multiple comparisons with the best (MCB) diagram (Demšar, 2006). In this setting, each algorithm is assigned a rank on every dataset, ranks are averaged across datasets, and the MCB plot reports these mean ranks together with the critical distance (CD) at a prescribed confidence level. At the 5% level, any method whose mean rank differs from the best method by more than the CD is deemed significantly worse than the best under the Nemenyi correction.

Across the eight datasets and eight algorithms, the Friedman omnibus test rejects the null hypothesis of equal performance for both discrimination and calibration, with $p = 0.0021$ for Concordance and $p = 4 \times 10^{-4}$ for

| Dataset | KMNet-NLS | KMNet-NLE | KMNet-NPS | KMNet-NPE | KMNet-BLS | KMNet-BLE | KMNet-BPS | KMNet-BPE |
|---|---|---|---|---|---|---|---|---|
| Flchain | $\underline{0.099}$ ± 0.002 | 0.100 ± 0.003 | 0.099 ± 0.002 | 0.100 ± 0.002 | 0.100 ± 0.002 | **0.099** ± 0.002 | 0.099 ± 0.002 | 0.099 ± 0.002 |
| Metabric | 0.163 ± 0.006 | 0.164 ± 0.009 | **0.160** ± 0.006 | 0.163 ± 0.004 | 0.164 ± 0.010 | $\underline{0.160}$ ± 0.004 | 0.161 ± 0.007 | 0.162 ± 0.004 |
| Nacd | 0.146 ± 0.005 | 0.146 ± 0.005 | 0.146 ± 0.007 | $\underline{0.143}$ ± 0.004 | 0.144 ± 0.002 | 0.147 ± 0.006 | 0.147 ± 0.007 | **0.142** ± 0.004 |
| Nacd-Col | 0.178 ± 0.017 | $\underline{0.170}$ ± 0.024 | 0.184 ± 0.013 | 0.173 ± 0.017 | 0.183 ± 0.029 | **0.170** ± 0.018 | 0.175 ± 0.010 | 0.173 ± 0.005 |
| Nwtco | 0.089 ± 0.002 | 0.088 ± 0.005 | 0.092 ± 0.006 | 0.087 ± 0.004 | **0.084** ± 0.006 | $\underline{0.086}$ ± 0.001 | 0.087 ± 0.006 | 0.086 ± 0.005 |
| Recidivism | 0.180 ± 0.017 | 0.179 ± 0.002 | 0.177 ± 0.010 | **0.173** ± 0.001 | $\underline{0.173}$ ± 0.004 | 0.176 ± 0.007 | 0.177 ± 0.006 | 0.177 ± 0.005 |
| Rotterdam-Gbsg | 0.178 ± 0.007 | $\underline{0.167}$ ± 0.001 | 0.167 ± 0.007 | 0.170 ± 0.000 | 0.173 ± 0.002 | 0.168 ± 0.004 | 0.171 ± 0.000 | **0.165** ± 0.002 |
| Support | 0.191 ± 0.002 | 0.192 ± 0.001 | 0.192 ± 0.001 | **0.191** ± 0.002 | 0.193 ± 0.002 | 0.191 ± 0.001 | 0.192 ± 0.003 | $\underline{0.191}$ ± 0.001 |

Table 11: Integrated Brier Score (Lower is Better) (mean ± std)

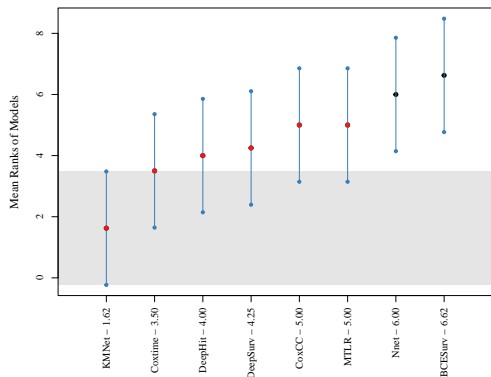

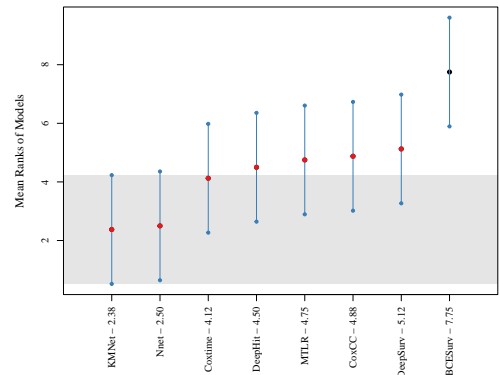

(a) MCB plot for Concordance (mean ranks; lower is better).

(b) MCB plot for IBS (mean ranks; lower is better).

Figure 4: Nemenyi post hoc analysis after the Friedman test across all datasets. The horizontal bar indicates the critical distance (CD) at the 5% level; methods whose mean ranks differ by less than the CD are not statistically distinguishable under the Nemenyi correction.

IBS (Friedman, 1937; 1940). The corresponding Nemenyi critical distance is CD = 3.712. For Concordance, KMNet achieves the best mean rank (1.62); the differences in mean rank between KMNet and BCESurv (6.62), as well as between KMNet and NNet (6.00), exceed the CD, indicating that these two baselines are significantly worse than KMNet in terms of concordance at the 5% level (Nemenyi, 1963). For IBS, KMNet again attains the best mean rank (2.38); BCESurv has the worst mean rank (7.75) and lies beyond the CD from KMNet, implying significantly worse calibration for BCESurv relative to KMNet. The remaining methods fall within the CD from KMNet and are therefore not significantly different from KMNet under the Nemenyi correction.

To complement the rank-based Nemenyi analysis, we additionally perform paired Wilcoxon signed-rank tests across datasets, using KMNet as the reference method (see Table 12) (Wilcoxon, 1992). For Concordance, KMNet shows statistically significant improvements over BCESurv, CoxCC, MTLR, and NNet after FDR correction (all adjusted $p \leq 0.025$), while the differences to CoxTime and DeepHit are not significant (adjusted $p = 0.1415$). The comparison with DeepSurv is significant before adjustment ($p = 0.0423$) but not after FDR correction ($p = 0.0592$), indicating a weaker and less consistent advantage across baselines. For IBS, KMNet significantly outperforms BCESurv, CoxCC, and MTLR after FDR correction (all adjusted $p = 0.0487$), whereas differences to CoxTime, DeepHit, and DeepSurv are not significant at the 5% level. Notably, the IBS comparison against NNet is not significant ($p = 1.0$), suggesting comparable calibration between these two methods under our evaluation protocol. To provide an overall summary, we additionally report Fisher-combined $p$-values (Fisher, 1970). The combined tests are statistically significant for both concordance and IBS, providing aggregate evidence of performance differences across datasets. However, after FDR correction, the pairwise comparisons are significant against only four of the seven baselines for

Table 12: Paired Wilcoxon signed-rank tests comparing each baseline to KMNet across datasets. We report two-sided p-values and FDR-adjusted p-values for both discrimination (Concordance) and calibration (IBS).

| Method | Concordance | | IBS | |
|---|---|---|---|---|
| | $p$ | $p$ (FDR) | $p$ | $p$ (FDR) |
| BCESurv | 0.0143 | 0.0250 | 0.0143 | 0.0487 |
| CoxCC | 0.0143 | 0.0250 | 0.0209 | 0.0487 |
| CoxTime | 0.1415 | 0.1415 | 0.1415 | 0.1651 |
| DeepHit | 0.1415 | 0.1415 | 0.0587 | 0.0822 |
| DeepSurv | 0.0423 | 0.0592 | 0.0587 | 0.0822 |
| MTLR | 0.0143 | 0.0250 | 0.0143 | 0.0487 |
| NNet | 0.0143 | 0.0250 | 1.0000 | 1.0000 |
| Fisher combined $p$-value | $< 10^{-4}$ | $< 10^{-4}$ | $0.3 \times 10^{-3}$ | 0.0044 |

concordance and three of the seven baselines for IBS. Therefore, the favourable mean ranks and significant aggregate $p$-values support KMNet as a competitive overall method, but do not imply that it consistently and significantly outperforms every other baseline.

## 7 Simulation study

This section presents a controlled simulation study designed to complement the real-data experiments and to provide additional insight into the behaviour of KMNet under known data-generating mechanisms. Across Experiments, we use the same data-generation protocol so that observed differences are attributable to the model configuration or the specific analysis being performed rather than changes in the underlying data.

For each simulated individual $i = 1, \ldots, n$, we sample a feature vector $X_i \in \mathbb{R}^p$ with independent standard normal entries. Event times are generated from a proportional hazards mechanism with a log-linear risk score $\eta_i = X_i^\top \beta$, where $\beta$ has only a small number of non-zero coefficients. Specifically, the true event time $T_i$ is sampled from an exponential distribution with rate $\lambda_i = \lambda_0 \exp(\eta_i)$, where $\lambda_0 > 0$ is a baseline rate. Censoring times $C_i$ are sampled independently from an exponential distribution, with the rate chosen by bisection to match a prescribed censoring fraction. The observed time and event indicator are then given by

$$Y_i = \min\{T_i, C_i\}, \qquad \delta_i = 1\{T_i \leq C_i\}.$$

For each run, we split the simulated data into 80% development and 20% test sets. From the development set, we further hold out 10% as a validation set used for model selection when required. Continuous features are standardized using statistics computed on the training split only. To train discrete-time models, the observed times are discretized into $J$ intervals using the training split, and the same discretization is applied to validation and test data.

**Effect of $\alpha$ and $\lambda$** Figure 5 summarizes a grid-based sensitivity analysis on the synthetic dataset, where we vary the base-loss weight $\alpha$ and the ranking weight $\lambda$ while keeping the remaining KMNet configuration fixed. For each $(\alpha, \lambda)$ pair, we train KMNet on the training split and evaluate the time-dependent concordance and the integrated Brier score (IBS) on the validation split. The upper heatmap reports the validation concordance (higher is better), and the lower heatmap reports the validation IBS (lower is better).

Two complementary trends emerge. First, the best discrimination is achieved for moderate ranking strength and a non-negligible contribution from the ranking term, with the highest concordance attained at an intermediate $\lambda$ (here $\lambda = 0.5$) and a large $\alpha$ (here $\alpha = 0.8$). Second, the best calibration is achieved when the objective is dominated by the base BCE term, with the lowest IBS occurring at a large $\alpha$ (here $\alpha = 0.95$) and a smaller ranking weight (here $\lambda = 0.25$). These results illustrate the expected calibration–discrimination trade-off induced by the ranking component. Increasing $\lambda$ typically improves ordering in settings where individuals are comparable, but can degrade probability calibration when over-emphasized; conversely, larger $\alpha$ stabilizes probability estimation through the base BCE term, improving IBS but potentially limiting gains

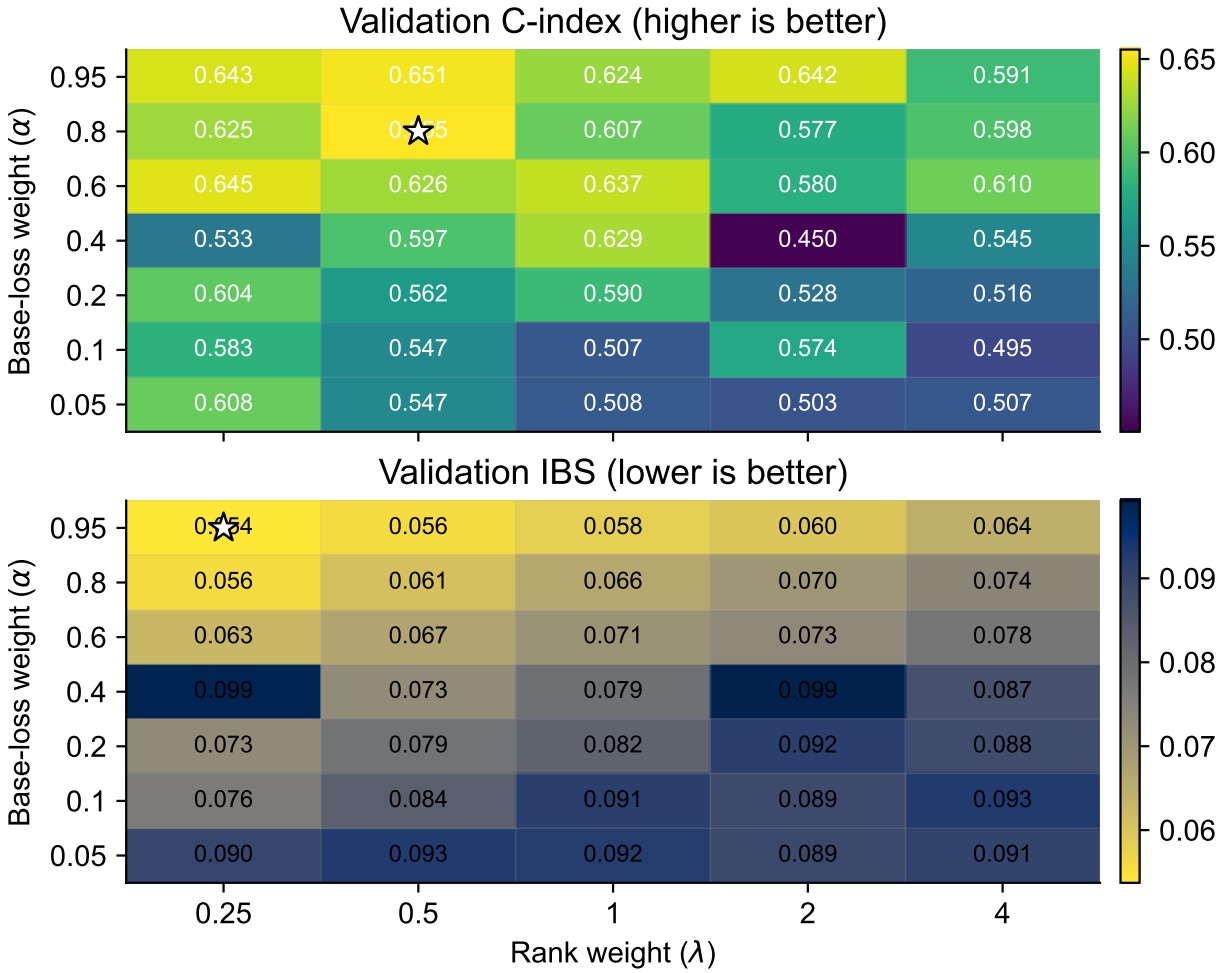

Figure 5: Sensitivity of KMNet to the loss weights on simulated data. Top: validation concordance (C-index in Figure). Bottom: validation IBS. Stars indicate the best configuration for each metric on this grid.

in concordance. Overall, the heatmaps indicate that KMNet is most effective in the regime of large $\alpha$ with moderate $\lambda$, motivating our use of Bayesian optimization to select $(\alpha, \lambda)$ rather than fixing them *a priori*.

**Sensitivity to the number of intervals $J$**  Discrete-time survival models depend on a user-specified discretization of the time axis into $J$ intervals. While a finer discretization can represent more detailed temporal dynamics, it also increases the number of outputs and may amplify optimization noise. In Experiment C, we assess the robustness of KMNet to the discretization resolution by varying the number of intervals $J \in \{32, 64, 128\}$ on the same simulated dataset and under the same training protocol. For each value of $J$, we discretize observed times using the training split, train KMNet with the proposed configuration, and report the time-dependent concordance and the integrated Brier score (IBS) on the validation split, averaged over multiple random seeds.

Figure 6 shows that KMNet is not overly sensitive to the choice of $J$ in this range. As $J$ increases, concordance tends to improve, while IBS may slightly worsen, indicating a modest discrimination–calibration trade-off. Overall, an intermediate resolution provides a favourable balance, with little additional benefit from increasing $J$ beyond this point, while larger $J$ increases the output dimensionality and training cost. This experiment supports the use of a mid-range discretization (e.g., $J = 64$) as a robust default that yields competitive performance without requiring careful dataset-specific tuning.

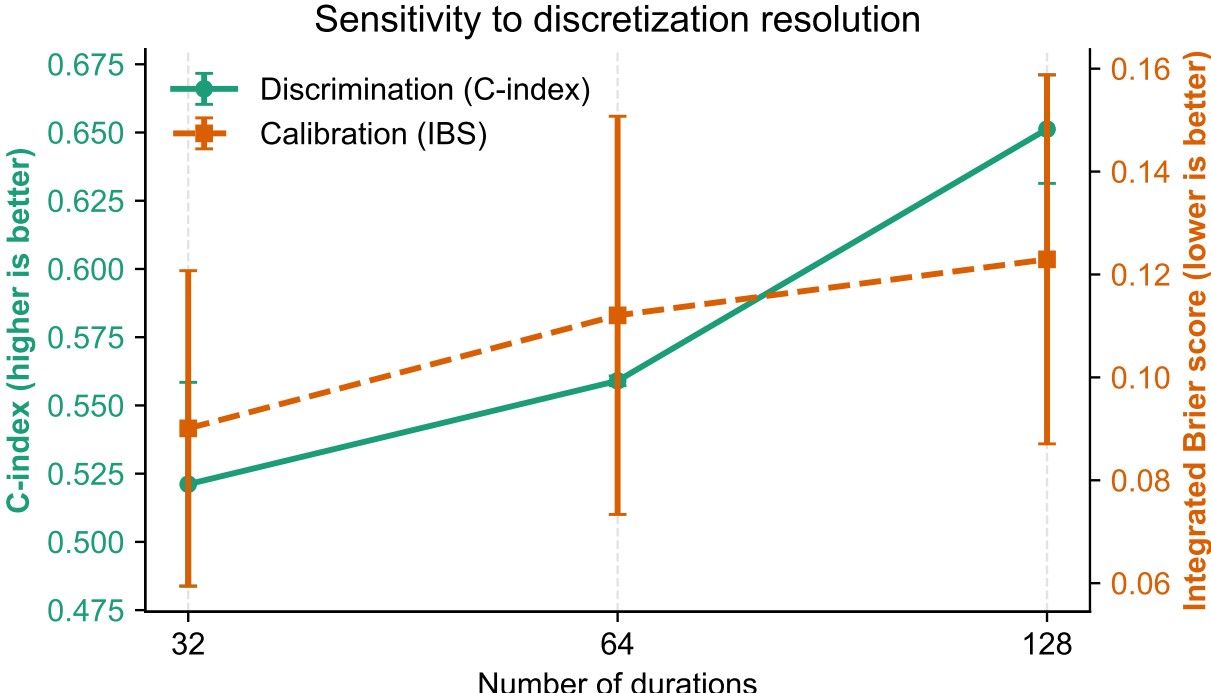

Figure 6: Sensitivity of KMNet to the discretization resolution on simulated data. We vary the number of discrete time intervals $J \in \{32, 64, 128\}$ and report mean $\pm$ standard deviation across for concordance (C-index in Figure) and integrated Brier score over random seeds.

**Risk stratification.**   To assess whether KMNet induces clinically meaningful separation of individuals into risk strata, we summarize each subject by the predicted survival at a reference horizon $t^\star$ and define a scalar risk score $r(x) = -\log \widehat{S}(t^\star \mid x)$. We set $t^\star$ to the empirical median of the observed times, which provides a balanced horizon with a substantial fraction of individuals still at risk. We then partition subjects into four equally sized groups according to risk quartiles and compute the Kaplan–Meier estimate within each group using the observed $(Y, \delta)$ pairs. The resulting curves, shown in Figure 7 exhibits clear separation and preserves the expected ordering from low to high risk, with the high-risk group showing the steepest early decline and the low-risk group maintaining the highest survival across follow-up. This stratification plot provides an interpretable complement to concordance and IBS by directly linking model predictions to empirical survival differences between risk groups. Risk stratification of real world dataset is deferred to Appendix D (see Figure 8)

## 8   Conclusion

We proposed Kaplan–Meier Net, a discrete-time neural survival model that learns interval-wise conditional survival probabilities and constructs patient-specific survival curves through a Kaplan–Meier style product. This design guarantees non-increasing survival predictions by construction while handling right censoring through an at-risk weighted training objective. We further introduced a conditional ranking loss computed at the event interval of the anchor individual, which directly targets discrimination in the conditional probabilities that define the survival curve. Across eight benchmark datasets, ~~KMNet achieved strong performance in both time-dependent concordance and integrated Brier score, with the best overall average rank on both metrics,~~ KMNet achieved competitive performance, with the best average rank for both concordance and IBS while always producing valid survival curves.

Several directions remain for future work. First, the current formulation uses a fixed time grid, and performance can depend on the choice of discretization. An adaptive or data-driven time grid, or multi-resolution grids that allocate more bins in high-event-density regions, may further improve both calibration and dis-

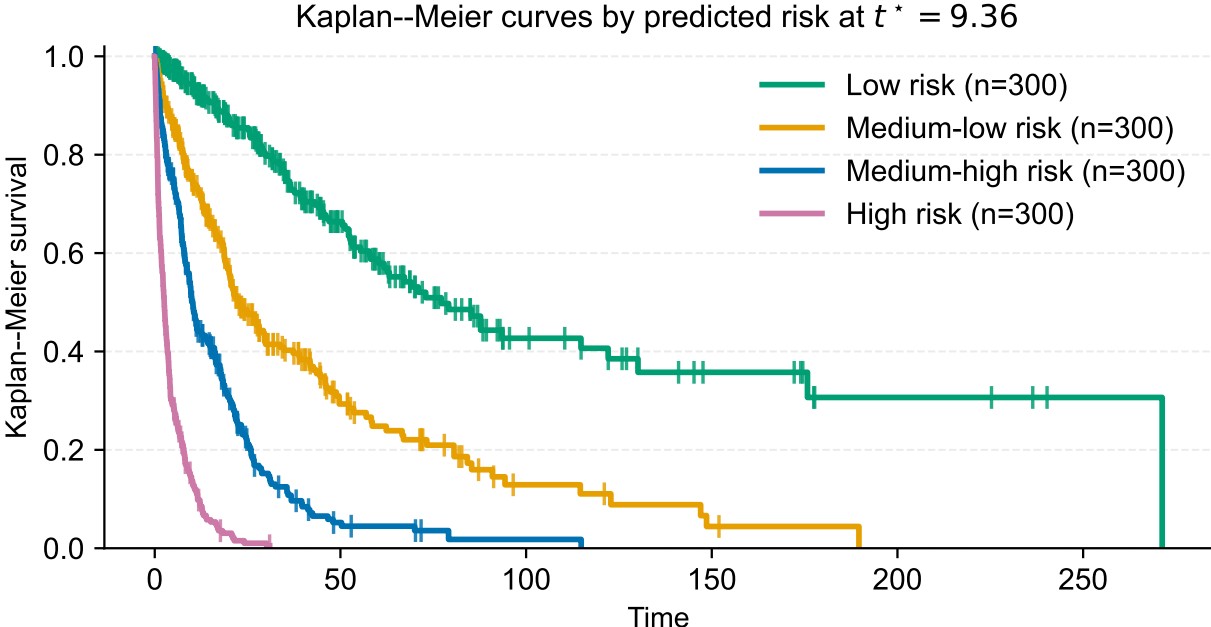

Figure 7: Kaplan–Meier curves for four groups formed by quartiles of the predicted risk $r(x) = -\log \widehat{S}(t^\star \mid x)$ at $t^\star = 9.36$ on simulated data.

crimination. Second, while our conditional ranking term improves risk stratification, it can be extended to incorporate time-varying comparisons across multiple horizons or to combine local and global ranking signals in a single objective. Third, extending KMNet to competing risks and multi-state settings would broaden its clinical applicability, since many real-world outcomes involve multiple event types. Fourth, further improvements in uncertainty quantification and calibration, such as conformal survival prediction or Bayesian treatments of the conditional probabilities, could provide more reliable decision support. Finally, integrating KMNet with interpretability tools and structured clinical priors, and evaluating it in prospective or external validation studies, are important steps toward deployment in high-stakes medical applications.

## Statement of Broader Impact

KMNet is a methodological contribution to discrete-time survival analysis that may support more reliable time-to-event risk estimation, prognosis, follow-up planning, and risk stratification by producing valid non-increasing survival curves and enabling evaluation of discrimination and calibration. However, the method is evaluated on retrospective medical and Recidivism benchmarks, and inaccurate or poorly calibrated predictions could contribute to inappropriate treatment decisions, unequal resource allocation, or harmful decisions about individuals. Monotonicity and strong benchmark performance do not establish clinical utility, causal validity, fairness, or deployment readiness, and performance may degrade under distribution shift or vary across subgroups. KMNet should therefore not be used as a stand-alone decision system in clinical, criminal-justice, or administrative settings. Any deployment would require external and prospective validation, horizon-specific and subgroup calibration assessment, fairness and distribution-shift analysis, evaluation of net benefit relative to existing practice, transparent communication of uncertainty, qualified human oversight, institutional and regulatory review, and appropriate safeguards for privacy, consent, data governance, and dataset provenance.

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

## Appendix

## A  Deephit Ranking Loss in terms of KMNet

DeepHit formulates pairwise ranking in terms of the cumulative incidence evaluated at the observed event time of an anchor individual (Lee et al., 2018). In the single-event setting considered here, the corresponding cumulative event probability at grid point $t_j$ is

$$\widehat{F}(t_j \mid x) = 1 - \widehat{S}(t_j \mid x) = 1 - \prod_{\ell=1}^{j} p_\ell(x).$$

To construct a controlled DeepHit-style comparator, we retain the same set of comparable pairs $\mathcal{C}_i$ and replace the proposed conditional-survival margin by the cumulative-risk margin $m_{ik}^{\mathrm{CDF}} = \widehat{F}(t_{\kappa_i} \mid X_i) - \widehat{F}(t_{\kappa_i} \mid X_k), \forall k \in \mathcal{C}_i$. Because individual $i$ experiences the event earlier than individual $k$, a correctly ordered pair should satisfy $\widehat{F}(t_{\kappa_i} \mid X_i) > \widehat{F}(t_{\kappa_i} \mid X_k)$. The corresponding ranking objective is

$$\mathcal{L}_{\mathrm{Rank}}^{\mathrm{CDF}} = \frac{1}{N} \sum_{i=1}^{N} 1\{\delta_i = 1\} 1\{|\mathcal{C}_i| > 0\} \frac{1}{|\mathcal{C}_i|} \sum_{k \in \mathcal{C}_i} \exp\left(-\frac{m_{ik}^{\mathrm{CDF}}}{\tau}\right).$$

Equivalently, since $\widehat{F} = 1 - \widehat{S}$, the cumulative-risk margin can be written as

$$m_{ik}^{\mathrm{CDF}} = \widehat{S}(t_{\kappa_i} \mid X_k) - \widehat{S}(t_{\kappa_i} \mid X_i) = \prod_{\ell=1}^{\kappa_i} p_\ell(X_k) - \prod_{\ell=1}^{\kappa_i} p_\ell(X_i).$$

The proposed KMNet ranking objective instead uses the local event-interval margin by $p_{\kappa_i}(X_k) - p_{\kappa_i}(X_i)$. Both objectives encourage the individual who survives longer to have lower cumulative event risk at the anchor time. However, they impose this ordering on different predictive quantities. The DeepHit-style margin compares cumulative event probabilities and therefore depends on all conditional survival outputs from the beginning of follow-up through interval $\kappa_i$. In contrast, the proposed margin compares only the conditional probability of surviving the anchor interval $(t_{\kappa_i - 1}, t_{\kappa_i}]$, given survival to its beginning. Consequently, KMNet applies ranking directly to the same interval-specific quantity that enters the product construction at the observed event interval, rather than to a cumulative quantity that aggregates information from all preceding intervals.

## B  Performance Metric

**Time Dependent Concordance**   The concordance index evaluates discrimination by measuring how well a model preserves the ordering of event times. It is particularly natural for proportional hazards models, where the relative risk ordering is consistent over time. For general non-proportional survival models, the choice of evaluation time matters, so we use a time-dependent concordance measure in the spirit of Antolini et al. (2005). In this formulation, a pair of individuals $(i, j)$ is comparable when individual $i$ experiences the event before individual $j$, that is $T_i < T_j$ with $\delta_i = 1$. The prediction is concordant if the model assigns lower survival to the individual who fails earlier when evaluated at $T_i$,

$$C = \Pr\left(\hat{S}(T_i \mid x_i) < \hat{S}(T_i \mid x_j) \,\Big|\, T_i < T_j, \ \delta_i = 1\right). \tag{1}$$

**Integrated Brier Score**   The Brier score assesses the accuracy of probabilistic predictions through a mean squared error criterion. In binary classification, for labels $y_i \in \{0, 1\}$ with predicted probabilities $p_i$, it is $BS = \frac{1}{N} \sum_i (y_i - p_i)^2$. In survival analysis, one fixes a time $t$ and forms the binary outcome $1\{T_i > t\}$ (whether the event occurs after $t$), comparing it to the predicted survival probability $\hat{S}(t \mid x_i)$. To account

for right censoring, we use the inverse-probability-of-censoring weighting (IPCW) estimator of Graf et al. (1999),

$$BS(t) = \frac{1}{N} \sum_{i=1}^{N} \left[ \frac{\hat{S}(t \mid x_i)^2 \, 1\{T_i \leq t, \ \delta_i = 1\}}{\hat{G}(T_i)} + \frac{\left(1 - \hat{S}(t \mid x_i)\right)^2 1\{T_i > t\}}{\hat{G}(t)} \right], \tag{2}$$

where $T_i$ is the observed time and $\delta_i$ the event indicator. The censoring weights use $\hat{G}$, the Kaplan–Meier estimate of the censoring survival function $G(t) = \Pr(C > t)$ obtained from the observed times with the event indicator reversed (the `censor_surv='km'` option of `pycox`'s `EvalSurv` (Kvamme et al., 2019)). Uncensored subjects that have already failed by $t$ are weighted by $1/\hat{G}(T_i)$ and subjects still at risk by $1/\hat{G}(t)$; subjects censored before $t$ contribute zero, their information being carried by the weights.

We summarize performance across time by the integrated Brier score. Rather than fixing an arbitrary constant horizon, we integrate over the full support of the observed times in each test set: the upper limit (horizon) is $\tau = \max_i T_i$ and the lower limit is $t_{\min} = \min_i T_i$. The integral is evaluated on a uniform grid of $M = 100$ points $t_{\min} = t_1 < \cdots < t_M = \tau$ by the trapezoidal rule and normalized by the grid width,

$$IBS = \frac{1}{\tau - t_{\min}} \int_{t_{\min}}^{\tau} BS(t) \, dt \ \approx \ \frac{1}{\tau - t_{\min}} \sum_{k=1}^{M-1} \frac{BS(t_k) + BS(t_{k+1})}{2} \, (t_{k+1} - t_k), \tag{3}$$

so that $IBS$ is a time-averaged Brier score on the Brier scale. All quantities are computed with `pycox`'s `EvalSurv`, and the *same* grid and censoring estimate $\hat{G}$ are used for every method on a given split, so the scores are directly comparable. Because $\tau$ is the largest observed time, it is dataset- and split-specific (e.g. $\tau \approx 330$ for Metabric, $\tau \approx 84$ for Nacd, and $\tau \approx 6143$ for Nwtco).

**D-calibration** While the (integrated) Brier score measures sharpness and accuracy, it does not directly test whether the predicted survival distributions are statistically consistent with the observed outcomes. For this we use the distributional calibration (D-calibration) test of Haider et al. (2020). The key object is the probability integral transform of each subject's observed time under its own predicted survival function,

$$u_i \ = \ \hat{S}(T_i \mid x_i), \tag{4}$$

i.e. the predicted probability of surviving beyond the observed time $T_i$. If the predicted distributions are correct, then for an uncensored subject $u_i$ is a draw from a Uniform$(0, 1)$ distribution. D-calibration tests exactly this uniformity.

We partition $[0, 1]$ into $B$ equal-width bins $I_b = \left[\frac{b-1}{B}, \frac{b}{B}\right)$, $b = 1, \ldots, B$, and accumulate a (soft) count $O_b$ in each bin. An *uncensored* subject ($\delta_i = 1$) contributes a full unit to the single bin containing $u_i$. A *censored* subject ($\delta_i = 0$) is only known to have survived beyond $C_i$, so its transformed value is known only to lie in $[0, u_i]$ with $u_i = \hat{S}(C_i \mid x_i)$; following Haider et al. (2020) we spread its unit of mass uniformly over that interval. Concretely, with $k = \lfloor B \, u_i \rfloor$ the bin containing $u_i$, the subject contributes

$$\frac{1}{B \, u_i} \text{ to each bin } b < k, \qquad \frac{u_i - (k/B)}{u_i} \text{ to bin } k, \tag{5}$$

which distributes a total mass of 1 over $[0, u_i]$ (a subject with $u_i \leq 0$ is placed entirely in the first bin). Summing over all $N$ subjects yields the observed bin masses $O_1, \ldots, O_B$ with $\sum_b O_b = N$.

Under the null hypothesis of D-calibration the transformed values are uniform, so each bin has expected mass $E_b = N/B$. We measure the discrepancy with Pearson's $\chi^2$ goodness-of-fit statistic

$$\chi^2 \ = \ \sum_{b=1}^{B} \frac{\left(O_b - E_b\right)^2}{E_b}, \qquad E_b = \frac{N}{B}, \tag{6}$$

which under the null follows a $\chi^2$ distribution with $B - 1$ degrees of freedom. The reported D-calibration $p$-value is the corresponding upper-tail probability $p = \Pr\left(\chi_{B-1}^2 \geq \chi^2\right)$. A large $p$-value ($p > 0.05$) means the uniformity hypothesis is *not* rejected at the 5% level, i.e. the model is D-calibrated; a small $p$-value indicates mis-calibration. The associated *D-calibration histogram* simply plots the bin masses $O_b$ against the uniform reference $E_b = N/B$, so a flat histogram corresponds to a well-calibrated model.

| Model | Hyperparameter | Search Space |
|---|---|---|
| Neural Structure (For all models) | Number of Layers | [1, 2, 4] |
| | Number of Nodes | [16, 32, 64] |
| | Learning Rate | [0.0001, 0.1] |
| CoxTime, NNet, BCESurv | J | [32, 64, 128] |
| ~~DeepSurv~~ | ~~Weight Decay~~ | ~~[0.01, 0.1]~~ |
| DeepHit | $\alpha$ | ~~[0.1, 0.5, 0.9]~~ (1e-4,1.0) |
| | $\sigma$ | (0.1, 10) |
| | J | [32, 64, 128] |
| | ~~Weight Decay~~ | ~~[0.01, 0.1]~~ |
| KMNet | $\alpha$ | ~~[0.1, 0.5, 0.9]~~ (1e-4,1.0) |
| | $\tau$ | (0.1, 10) |
| | $\lambda$ | Auto $\cup$ (0.25, 4) |
| | J | [32, 64, 128] |

Table 13: Hyperparameter search space for different model, Neural architecture search space is same for all the models

## C  Bayesian Optimization

We tune hyperparameters using Bayesian optimization, which treats the validation performance as a black-box function of a hyperparameter vector $\theta$ defined over a search space $\Theta$. Instead of exhaustively evaluating configurations on a fixed grid, Bayesian optimization uses the outcomes of previously evaluated trials to guide the selection of the next configuration, thereby concentrating computation on regions of the search space that are most likely to improve performance Snoek et al. (2012). This is particularly well suited to neural survival models, where each evaluation of $f(\theta)$ requires training a model and the cost of naive search quickly becomes prohibitive. In our study, the objective value $f(\theta)$ is computed on the held-out validation split after training the model with hyperparameters $\theta$ on the training split, following the data splitting protocol in Section 4.2. We use the Tree-structured Parzen Estimator Bergstra et al. (2011) (TPE) method, which is a Bayesian optimization strategy designed for mixed discrete and continuous hyperparameter spaces. In our experiments, each trial consists of selecting $\theta$ from the search space in Table 13, training the model on the training portion of the development split, and evaluating $f(\theta)$ on the validation portion. After completing the trial budget, the configuration with the best validation objective is selected and used for final model fitting and test evaluation. The complete tuning procedure is summarized in Algorithm 1.

## D  Experiment Results Continued

This section provides supplementary empirical analyses that extend the aggregate concordance and integrated Brier score comparisons reported in the main text. We examine complementary aspects of predictive performance, including horizon-specific calibration intercepts and slopes, median survival time error, real-cohort risk stratification, reliability, time-dependent Brier score, and time-dependent AUC. Together, these results provide a more detailed view of discrimination, calibration, and prognostic usefulness across datasets and evaluation horizons. The calibration-oriented analyses use the IBS-optimized model configurations, whereas discrimination-oriented analyses use the concordance-optimized configurations.

### D.1  D-calibration of postprocessed BCESurv

Table 14 reports the D-calibration of posprocessed BCESurv survival curve.

### D.2  Calibration Slope and Intercept

In this Section, We assess calibration-in-the-large (intercept) and calibration slope, the standard components of the calibration hierarchy (Van Calster et al., 2016; Cox, 1958), evaluated at horizon-specific risks as in survival calibration (Van Houwelingen, 2000).

---

**Algorithm 1** TPE-based Bayesian optimization used for hyperparameter tuning

---

**Input**
Training data $\mathcal{D}_{tr}$, validation data $\mathcal{D}_{va}$
Search space $\Theta$
Trial budget $B$
Quantile level $\gamma \in (0, 1)$ used to define good trials

**Output**
Best hyperparameters $\theta^\star$

1: Initialize an empty history $\mathcal{H} \leftarrow \emptyset$
2: **for** $b = 1$ to $B$ **do**
3:     **if** $|\mathcal{H}|$ is small **then**
4:         Sample $\theta_b$ from $\Theta$ using the prior distribution
5:     **else**
6:         Let $\{(\theta_k, f_k)\}_{k=1}^{|\mathcal{H}|} = \mathcal{H}$ where $f_k$ is the validation objective
7:         Set a threshold $f^\star$ as the $\gamma$-quantile of $\{f_k\}$
8:         Fit two TPE density models
9:         $l(\theta) \approx p(\theta \mid f(\theta) \leq f^\star)$ using trials with $f_k \leq f^\star$
10:        $g(\theta) \approx p(\theta \mid f(\theta) > f^\star)$ using trials with $f_k > f^\star$
11:        Draw candidates $\{\tilde{\theta}\}$ from $l(\theta)$ and select
12:        $\theta_b \in \arg\max_{\tilde{\theta}} l(\tilde{\theta})/g(\tilde{\theta})$
13:     **end if**
14:     Train the model with hyperparameters $\theta_b$ on $\mathcal{D}_{tr}$
15:     Compute validation objective $f_b = f(\theta_b)$ on $\mathcal{D}_{va}$
16:     Update history $\mathcal{H} \leftarrow \mathcal{H} \cup \{(\theta_b, f_b)\}$
17: **end for**
18: Return $\theta^\star \in \arg\min_{(\theta, f) \in \mathcal{H}} f$

---

| Dataset | BCE+`cummin` | BCE+`isotonic` |
|---|---|---|
| Flchain | $<0.001$ | $<0.001$ |
| Metabric | $<0.001$ | $<0.001$ |
| Nacd | $0.008$ | $0.025$ |
| Nacd-Col | $<0.001$ | $<0.001$ |
| Nwtco | $<0.001$ | $<0.001$ |
| Recidivism | $<0.001$ | $<0.001$ |
| Rotterdam-Gbsg | **0.477** | **0.518** |
| Support | $<0.001$ | $<0.001$ |

Table 14: D-calibration $p$-values for BCESurv and its two post-hoc monotonizations. A variant is D-calibrated when $p > 0.05$ (**bold**); post-processing does not change the outcome.

Table 15 reports horizon-specific calibration slopes, for which a value of 1 indicates ideal calibration. Across the 24 dataset–horizon combinations, KMNet attains an excellent or good calibration slope in 14 cases, the highest count among the compared methods, and is classified as poor or strongly miscalibrated in six cases, the lowest count. Its slopes are generally closer to the ideal value at the middle and later horizons, although substantial deviations remain for some dataset–horizon combinations, particularly at the first horizon of Nwtco, Recidivism, and Rotterdam-Gbsg, as well as the earlier horizons of Nacd-Col. These results provide descriptive evidence of comparatively stable horizon-specific calibration, but they do not indicate uniformly good calibration across all datasets or establish clinical utility.

Table 16 reports calibration-in-the-large across datasets and evaluation horizons, where values closer to zero indicate better agreement between the average predicted and observed risks. The intercepts vary considerably across datasets, with the largest deviations generally occurring at Q1, where the limited number of early

| Dataset | Horizon | BCESurv | CoxCC | CoxTime | DeepHit | DeepSurv | NNet | MTLR | KMNet |
|---|---|---|---|---|---|---|---|---|---|
| Flchain | Q1 | 1.431 | 1.613 | 0.986 | 0.621 | 1.563 | 0.934 | 0.892 | 0.836 |
| | Q2 | 1.116 | 0.902 | 0.948 | 0.670 | 0.850 | 0.982 | 0.855 | 0.876 |
| | Q3 | 1.018 | 0.947 | 0.917 | 0.753 | 0.895 | 0.926 | 0.890 | 0.936 |
| Metabric | Q1 | 1.313 | 2.694 | 4.267 | 0.836 | 3.965 | 0.435 | 0.718 | 0.755 |
| | Q2 | 0.882 | 0.634 | 1.214 | 0.691 | 0.858 | 0.451 | 0.769 | 0.854 |
| | Q3 | 0.783 | 0.634 | 0.713 | 0.841 | 0.830 | 0.568 | 0.713 | 0.755 |
| Nacd | Q1 | 1.820 | 3.741 | 2.886 | 3.287 | 3.502 | 2.182 | 2.383 | 1.249 |
| | Q2 | 0.884 | 2.090 | 0.984 | 1.019 | 1.334 | 0.822 | 0.875 | 0.873 |
| | Q3 | 1.672 | 0.733 | 0.724 | 1.120 | 0.938 | 2.001 | 0.959 | 0.720 |
| Nacd-Col | Q1 | 3.139 | 3.200 | 3.801 | 3.461 | 2.914 | 2.751 | 1.546 | 2.021 |
| | Q2 | 2.343 | 0.477 | 0.843 | 2.432 | 2.082 | 0.499 | 0.330 | 0.492 |
| | Q3 | 1.733 | 0.662 | 1.054 | 2.307 | 2.081 | 0.680 | 0.528 | 0.780 |
| Nwtco | Q1 | 1.689 | 2.434 | 2.973 | 1.442 | 2.239 | 3.107 | 1.831 | 4.613 |
| | Q2 | 1.054 | 2.138 | 1.693 | 1.876 | 1.356 | 1.041 | 2.146 | 2.104 |
| | Q3 | 1.062 | 1.089 | 1.685 | 0.752 | 1.085 | 1.123 | 1.886 | 1.298 |
| Recidivism | Q1 | 2.263 | 3.559 | 3.408 | 2.244 | 3.603 | 1.805 | 1.876 | 3.935 |
| | Q2 | 1.086 | 1.080 | 0.886 | 0.894 | 1.197 | 0.958 | 0.554 | 1.249 |
| | Q3 | 0.903 | 0.775 | 0.617 | 0.599 | 0.877 | 0.564 | 0.403 | 0.979 |
| Rotterdam-Gbsg | Q1 | 0.774 | 1.232 | 3.301 | 1.922 | 1.112 | 1.195 | 2.268 | 2.487 |
| | Q2 | 0.718 | 0.801 | 0.941 | 0.709 | 0.834 | 0.869 | 0.771 | 0.833 |
| | Q3 | 0.813 | 0.796 | 0.955 | 0.722 | 0.843 | 0.951 | 0.844 | 0.949 |
| Support | Q1 | 0.819 | 0.648 | 0.657 | 0.781 | 0.535 | 0.992 | 1.331 | 0.920 |
| | Q2 | 0.729 | 0.687 | 0.765 | 0.695 | 0.565 | 0.908 | 0.924 | 0.888 |
| | Q3 | 0.771 | 0.849 | 0.842 | 0.780 | 0.743 | 0.861 | 0.858 | 0.829 |
| *Calibration-category counts across 24 dataset–horizon combinations* | | | | | | | | | |
| Excellent | | 5 | 4 | 7 | 1 | 2 | 8 | 2 | 4 |
| Good | | 6 | 4 | 4 | 6 | 9 | 5 | 6 | 10 |
| Moderate | | 6 | 8 | 5 | 9 | 4 | 1 | 6 | 4 |
| Poor | | 4 | 2 | 2 | 3 | 4 | 5 | 5 | 3 |
| Strongly miscalibrated | | 3 | 6 | 6 | 5 | 5 | 5 | 5 | 3 |
| **Excellent or good ↑** | | 11 | 8 | 11 | 7 | 11 | 13 | 8 | **14** |
| **Poor or strongly miscalibrated ↓** | | 7 | 8 | 8 | 8 | 9 | 10 | 10 | **6** |

Table 15: Calibration slopes across datasets and evaluation horizons. The ideal calibration slope is 1. Here, Q1, Q2, and Q3 denote the (25%), (50%), and (75%) time horizons, respectively. Cell colours represent the multiplicative deviation $|\log(s)|$ from this ideal: dark green denotes excellent calibration, followed by light green, yellow, orange, and red as the deviation increases. The rows at the bottom report the number of dataset–horizon combinations assigned to each descriptive category, out of 24 combinations per model.

The calibration-slope categories are defined using $d(s) = |\log(s)|$, the multiplicative deviation from the ideal slope $s = 1$. This measure treats reciprocal deviations symmetrically; for example, $s = 0.5$ and $s = 2$ have equal deviations because $|\log(0.5)| = |\log(2)|$. We classify slopes as excellent when $d(s) \leq 0.10$, good when $0.10 < d(s) \leq 0.25$, moderate when $0.25 < d(s) \leq 0.50$, poor when $0.50 < d(s) \leq 0.75$, and strongly miscalibrated when $d(s) > 0.75$. These thresholds correspond, respectively, to $s \in [0.90, 1.11]$, $s \in [0.78, 0.90) \cup (1.11, 1.28]$, $s \in [0.61, 0.78) \cup (1.28, 1.65]$, $s \in [0.47, 0.61) \cup (1.65, 2.12]$, and $s < 0.47$ or $s > 2.12$.

events can make the estimates unstable. Compared with BCESurv, KMNet produces an intercept closer to zero in 14 of the 24 dataset–horizon combinations, including all three horizons on Metabric, Nacd, Nacd-Col, and Recidivism. Across all settings, KMNet achieves a lower mean absolute calibration intercept than BCESurv ((0.701) versus (0.859)). The improvement is particularly evident at Q3, where the mean absolute intercept decreases from (0.422) for BCESurv to (0.178) for KMNet. At Q2, the two methods are broadly comparable, with mean absolute intercepts of (0.306) for KMNet and (0.276) for BCESurv. Overall, these results indicate that KMNet provides competitive calibration-in-the-large and tends to yield better-centered risk predictions than BCESurv, particularly at the later evaluation horizon, although the advantage is not uniform across all datasets.

| Dataset | Horizon | BCESurv | CoxCC | CoxTime | DeepHit | DeepSurv | NNet | MTLR | KMNet |
|---|---|---|---|---|---|---|---|---|---|
| Flchain | Q1 | -1.172 | -0.690 | 0.134 | 0.428 | -0.691 | -0.002 | 0.211 | 0.085 |
| | Q2 | -0.476 | 0.275 | 0.241 | 0.425 | 0.270 | 0.091 | 0.242 | 0.191 |
| | Q3 | -0.005 | 0.175 | 0.167 | 0.302 | 0.162 | 0.108 | 0.134 | 0.065 |
| Metabric | Q1 | -1.015 | -0.706 | -1.044 | 0.183 | -0.888 | 0.384 | 0.143 | -0.361 |
| | Q2 | -0.225 | 0.208 | -0.032 | 0.123 | 0.115 | 0.232 | -0.109 | -0.173 |
| | Q3 | -0.384 | -0.016 | -0.127 | -0.309 | -0.096 | -0.265 | -0.272 | -0.210 |
| Nacd | Q1 | -2.523 | -2.782 | -2.429 | -3.052 | -2.152 | -0.893 | -1.494 | -1.230 |
| | Q2 | 0.176 | -0.790 | 0.498 | 0.196 | -0.003 | 0.592 | 0.383 | 0.023 |
| | Q3 | 0.962 | -0.224 | 0.070 | 0.276 | -0.102 | 1.138 | 0.167 | -0.353 |
| Nacd-Col | Q1 | -3.022 | -2.778 | -1.722 | -2.309 | -2.959 | -2.859 | -3.070 | -2.181 |
| | Q2 | -1.153 | 0.653 | 0.605 | -1.132 | -0.675 | 0.395 | 0.861 | 0.397 |
| | Q3 | -1.535 | 0.213 | 0.108 | -1.606 | -1.179 | -0.452 | 0.077 | 0.192 |
| Nwtco | Q1 | -3.357 | -2.154 | -3.098 | -1.670 | -2.384 | -3.737 | -3.683 | -4.828 |
| | Q2 | -0.020 | -0.750 | -0.800 | -0.510 | -0.959 | 0.123 | -1.465 | -1.048 |
| | Q3 | -0.125 | -0.018 | -0.969 | 0.343 | -0.182 | 0.021 | -0.437 | -0.132 |
| Recidivism | Q1 | -3.718 | -1.122 | -2.090 | -0.975 | -1.176 | -0.975 | -1.819 | -1.200 |
| | Q2 | -0.049 | 0.000 | 0.044 | 0.275 | -0.049 | 0.171 | 0.082 | 0.001 |
| | Q3 | 0.176 | 0.140 | 0.185 | 0.438 | 0.112 | 0.258 | 0.114 | 0.133 |
| Rotterdam-Gbsg | Q1 | 0.020 | -0.608 | -1.396 | -1.364 | -0.523 | -0.592 | -1.258 | -1.360 |
| | Q2 | 0.019 | -0.200 | -0.180 | -0.197 | -0.180 | -0.247 | -0.102 | -0.174 |
| | Q3 | -0.153 | -0.235 | -0.224 | -0.290 | -0.201 | -0.320 | -0.241 | -0.291 |
| Support | Q1 | 0.198 | 0.069 | 0.072 | 0.444 | 0.091 | 1.650 | 1.631 | 1.714 |
| | Q2 | 0.090 | 0.051 | 0.055 | 0.009 | 0.075 | 0.399 | 0.405 | 0.442 |
| | Q3 | 0.035 | -0.030 | -0.024 | -0.085 | -0.024 | -0.038 | -0.035 | 0.050 |

Table 16: Calibration intercept across datasets and evaluation horizons. Here, Q1, Q2, and Q3 denote the (25%), (50%), and (75%) time horizons, respectively. The ideal intercept is (0); positive values indicate systematic underestimation of risk, or equivalently overestimation of survival, whereas negative values indicate systematic overestimation of risk. Estimates at Q1 should be interpreted cautiously for datasets with few early events, as sparse event counts can produce unstable intercepts and extreme logits.

| Dataset | BCESurv | CoxCC | CoxTime | DeepHit | DeepSurv | NNet | MTLR | KMNet |
|---|---|---|---|---|---|---|---|---|
| Flchain | **1670.726** | 1966.129 | 2075.285 | 1770.378 | 1903.853 | 1960.186 | 1823.428 | 1747.447 |
| Metabric | 63.524 | 82.260 | 82.095 | 90.237 | 83.371 | 81.138 | 80.823 | **60.252** |
| Nacd | **10.347** | 11.670 | 11.027 | 12.211 | 12.256 | 11.094 | 11.210 | 12.106 |
| Nacd-Col | 13.871 | 24.443 | 18.232 | 19.758 | 20.173 | 23.789 | 16.459 | **12.120** |
| Nwtco | 2322.549 | 4710.318 | 5009.355 | 4724.871 | 4519.327 | 2948.110 | 4663.564 | **926.350** |
| Recidivism | 45.657 | 48.391 | 43.391 | **36.818** | 47.200 | 48.800 | 56.070 | 41.854 |
| Rotterdam-Gbsg | 23.419 | 25.374 | 24.881 | 27.473 | 26.742 | 29.917 | 25.017 | **22.044** |
| Support | 314.521 | 383.839 | 407.187 | 475.424 | 361.937 | 375.970 | 441.970 | **184.522** |

Table 17: Mean absolute error (MAE) of the predicted median survival time among individuals with an observed event. Lower values indicate more accurate median survival-time predictions, and the best result for each dataset is shown in **bold**.

### D.3  MAE Loss

Table 17 evaluates the accuracy of the predicted median survival time, an interpretable patient-level summary defined as the time at which the predicted survival probability first falls to or below 0.5. For the set of individuals with observed events, the metric is computed as

$$\text{MAE}_{\text{event}} = \frac{1}{N_{\text{event}}} \sum_{i:\delta_i=1} |\widehat{m}_i - T_i|,$$

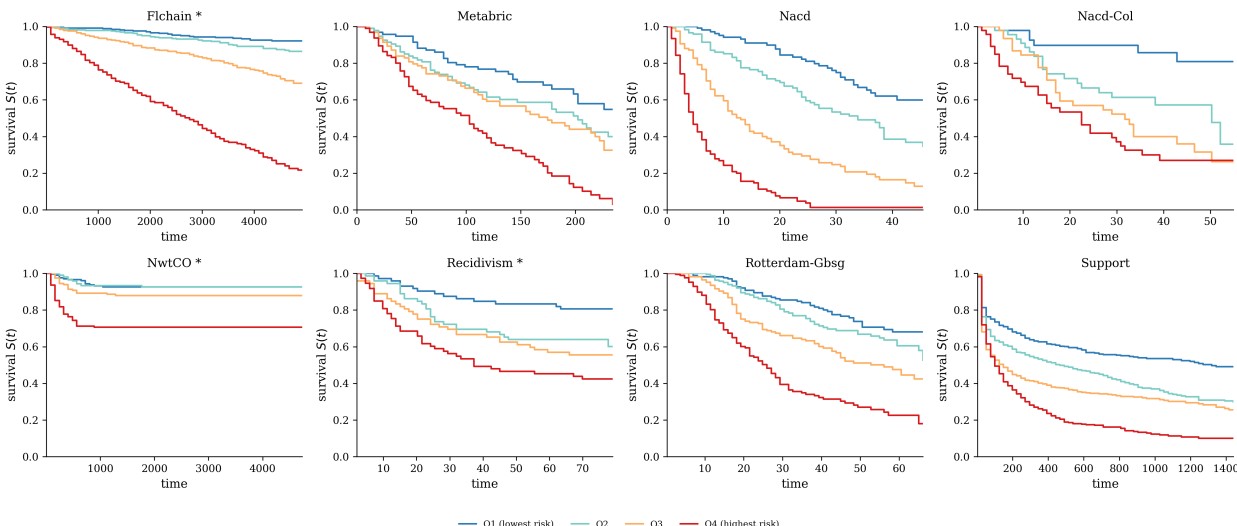

Figure 8: Real-cohort risk stratification for KMNet on all eight datasets. In each panel the test subjects are split into quartiles of the predicted risk $r(x) = -\log S(t^\star \mid x)$ evaluated at $t^\star$, the cohort median survival time (the time at which the pooled Kaplan–Meier curve reaches 0.5). Each coloured line is the Kaplan–Meier survival curve of one risk quartile computed on the real observed times and events: Q1 = lowest predicted risk, Q4 = highest. Well-ordered, well-separated curves indicate that the predicted risk correctly ranks subjects. Panels marked with an asterisk (∗) are datasets whose Kaplan–Meier curve never reaches 0.5 (heavy censoring), for which $t^\star$ falls back to the median observed event time.

where $\widehat{m}_i$ is the predicted median survival time, $T_i$ is the observed event time, and $\delta_i = 1$ indicates an uncensored observation. The metric is expressed in the original time units of each dataset and therefore provides a direct measure of the typical error in a clinically interpretable survival summary. KMNet achieves the lowest MAE on five of the eight datasets, including substantial reductions on Nwtco and Support, while BCESurv performs best on Flchain and Nacd and DeepHit performs best on Recidivism. These results indicate that KMNet's monotone survival curves often translate into accurate and unambiguous median survival-time estimates. Because censored individuals are excluded, however, this measure is descriptive of subjects with observed events and should be interpreted alongside the censoring-aware concordance, IBS, and calibration analyses.

## D.4 Risk Stratification on Real Data

## D.5 D-calibration Plots

## D.6 Reliability, Brier, and AUC over time horizon

Figures 17–19 provide time- and horizon-specific diagnostics that complement the aggregate IBS and concordance results. The reliability curves in Figure 17 show substantial overlap among the models, although agreement with the ideal diagonal varies across datasets and predicted-risk ranges. The curves for Nacd are comparatively close to the diagonal, whereas Nacd-Col exhibits greater bin-to-bin variability. Several models tend to lie below the diagonal at higher predicted risks on Flchain, Metabric, and Rotterdam-Gbsg, indicating some over-prediction of event risk, while portions of the Recidivism, Nwtco, and Support curves lie above the diagonal, indicating under-prediction. The time-dependent IPCW Brier-score curves in Figure 18 largely overlap over the central follow-up period, suggesting that differences in prediction error are generally modest and dataset dependent. More visible separation occurs late in follow-up for Nwtco and near the evaluation boundaries for Flchain and Recidivism, where the estimates may also be less stable because fewer individuals remain at risk and IPCW weights become more variable. Likewise, the time-dependent AUC curves in Figure 19 show broadly similar discrimination trajectories for most models, with no method uniformly dominating across all datasets and times (Uno et al., 2007; Heagerty & Zheng, 2005) Flchain

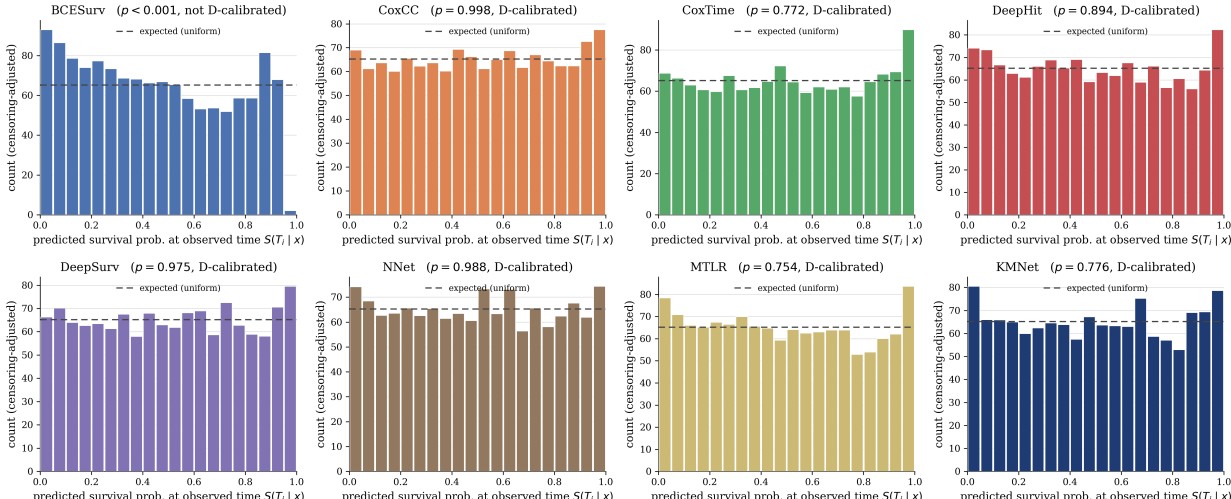

Figure 9: D-calibration histograms for all models on the Flchain dataset. Dashed line: uniform expected count; $p$ is the D-calibration test $p$-value.

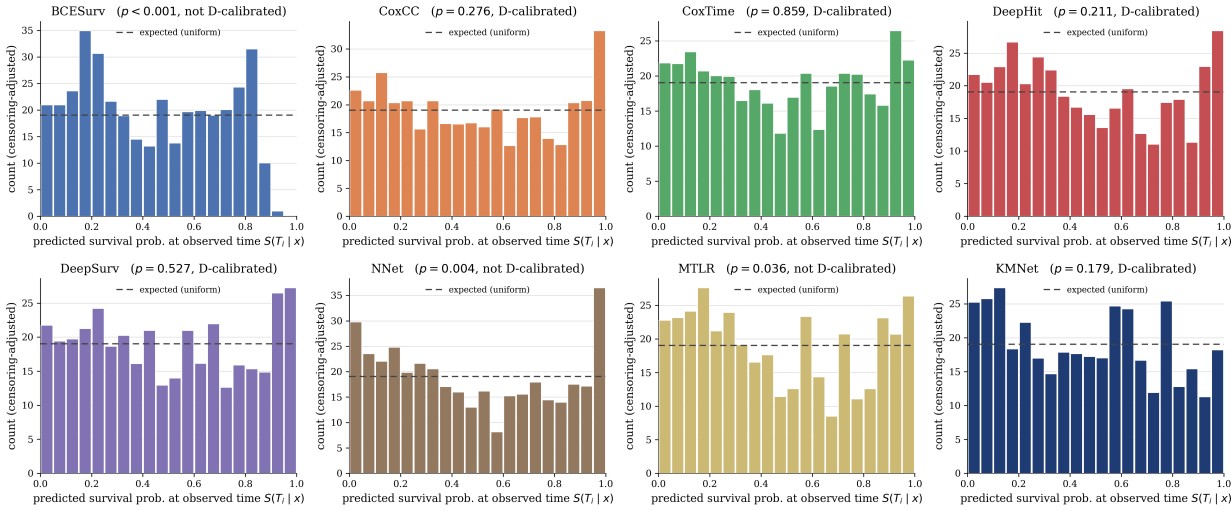

Figure 10: D-calibration histograms for all models on the Metabric dataset. Dashed line: uniform expected count; $p$ is the D-calibration test $p$-value.

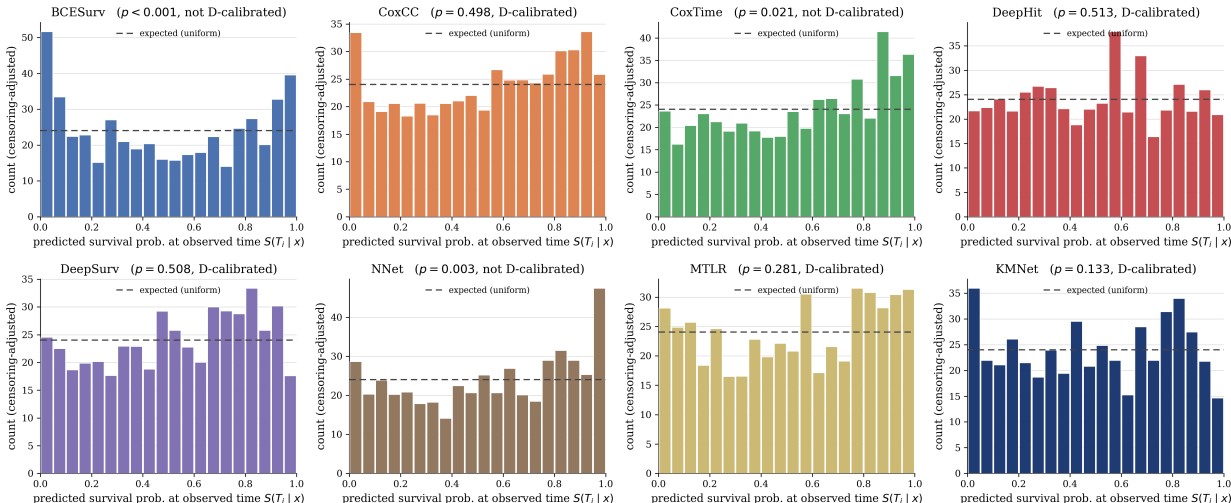

Figure 11: D-calibration histograms for all models on the Nacd dataset. Dashed line: uniform expected count; $p$ is the D-calibration test $p$-value.

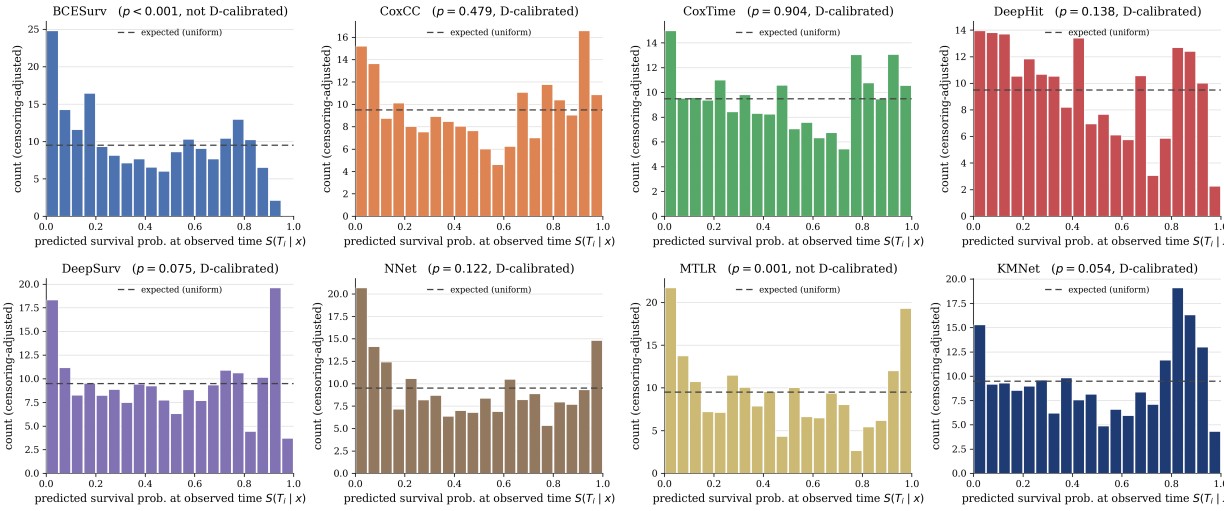

Figure 12: D-calibration histograms for all models on the Nacd-Col dataset. Dashed line: uniform expected count; $p$ is the D-calibration test $p$-value.

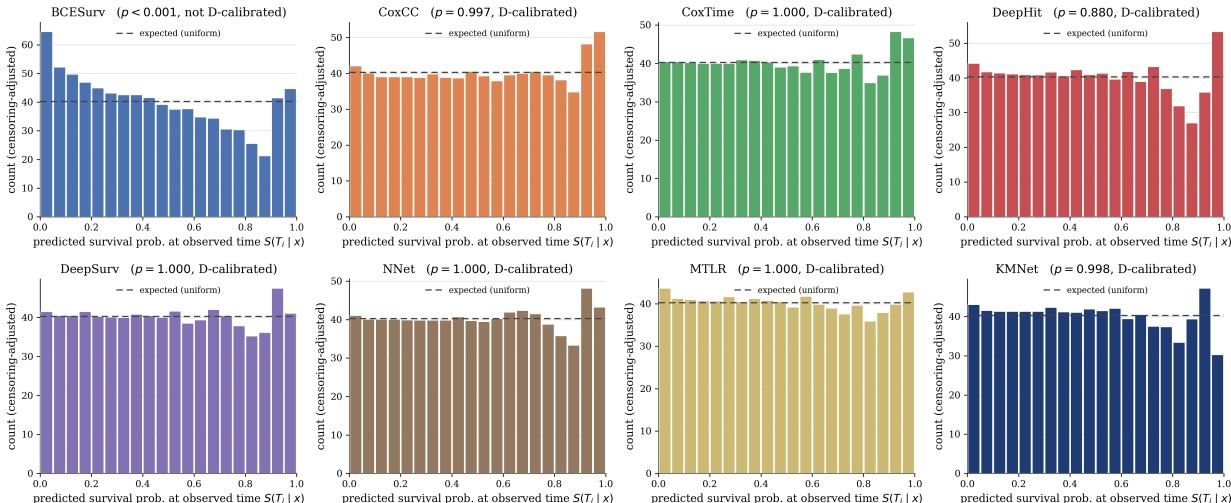

Figure 13: D-calibration histograms for all models on the Nwtco dataset. Dashed line: uniform expected count; $p$ is the D-calibration test $p$-value.

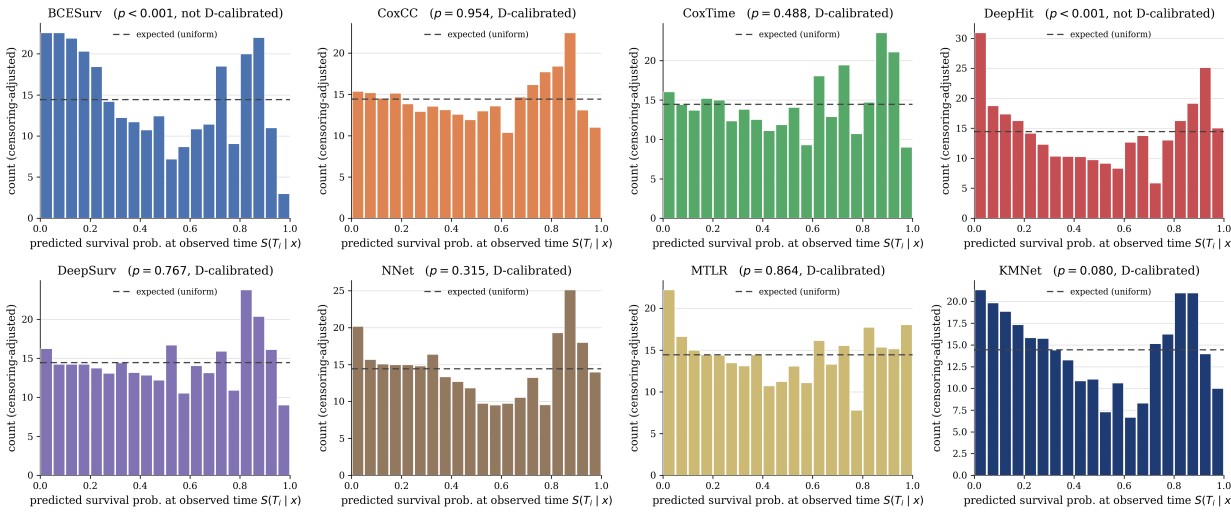

Figure 14: D-calibration histograms for all models on the Recidivism dataset. Dashed line: uniform expected count; $p$ is the D-calibration test $p$-value.

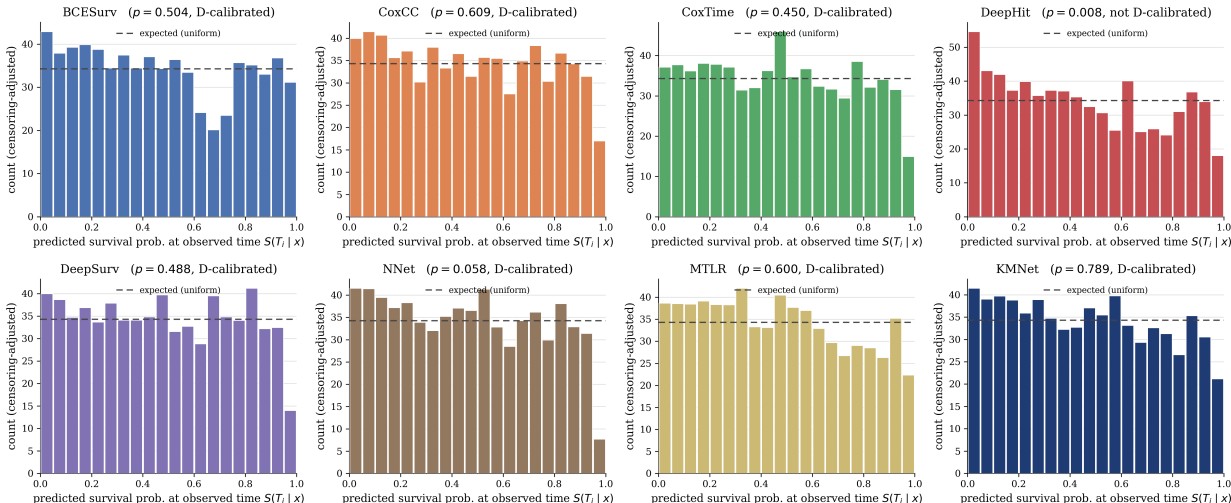

Figure 15: D-calibration histograms for all models on the ROTTERDAM/GBSG dataset. Dashed line: uniform expected count; $p$ is the D-calibration test $p$-value.

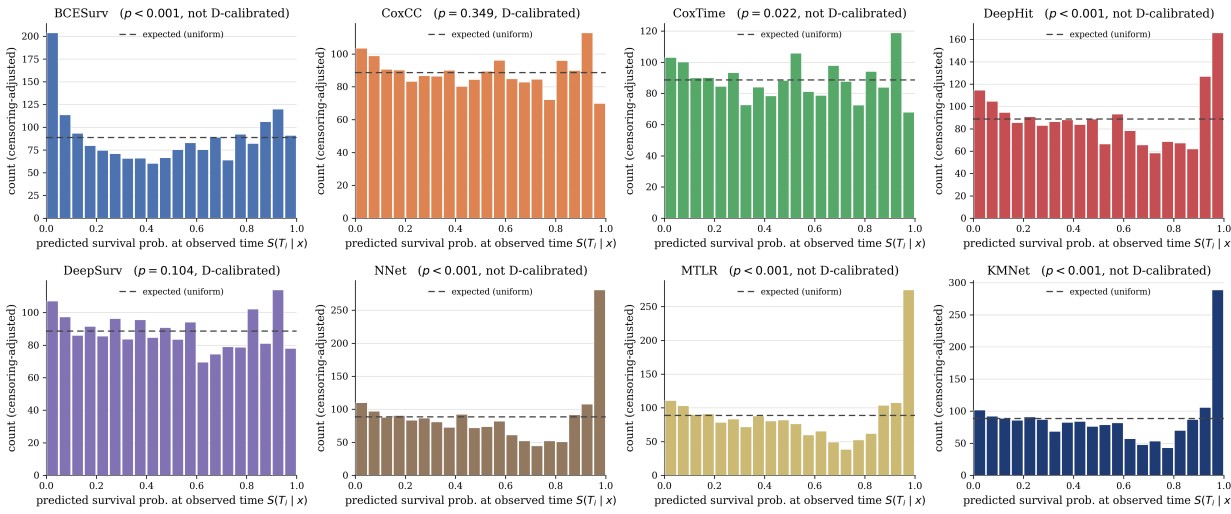

Figure 16: D-calibration histograms for all models on the Support dataset. Dashed line: uniform expected count; $p$ is the D-calibration test $p$-value.

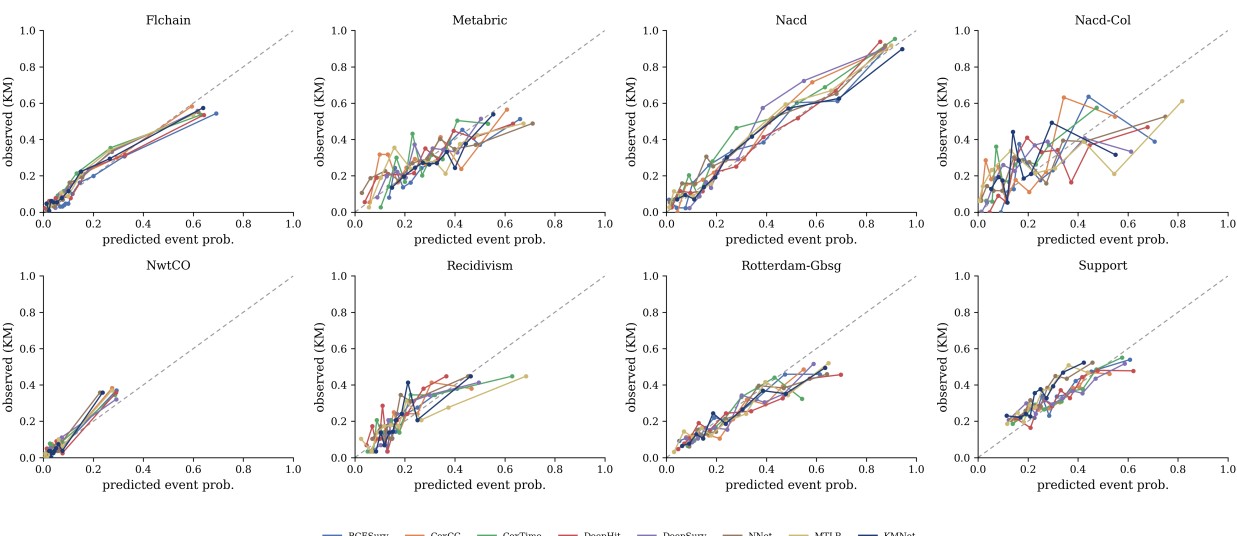

Figure 17: Reliability curves at the median (50%) event-time horizon for all eight models on each dataset. Within each panel, subjects are grouped into predicted-risk bins; the $x$-axis is the mean predicted event probability $1 - S(t^\star \mid x)$ and the $y$-axis the Kaplan–Meier observed event probability in that bin. The dashed diagonal is perfect calibration: points above it indicate under-prediction of risk, points below over-prediction. The curves exhibit substantial overlap, although departures from the diagonal vary across datasets and predicted-risk ranges.

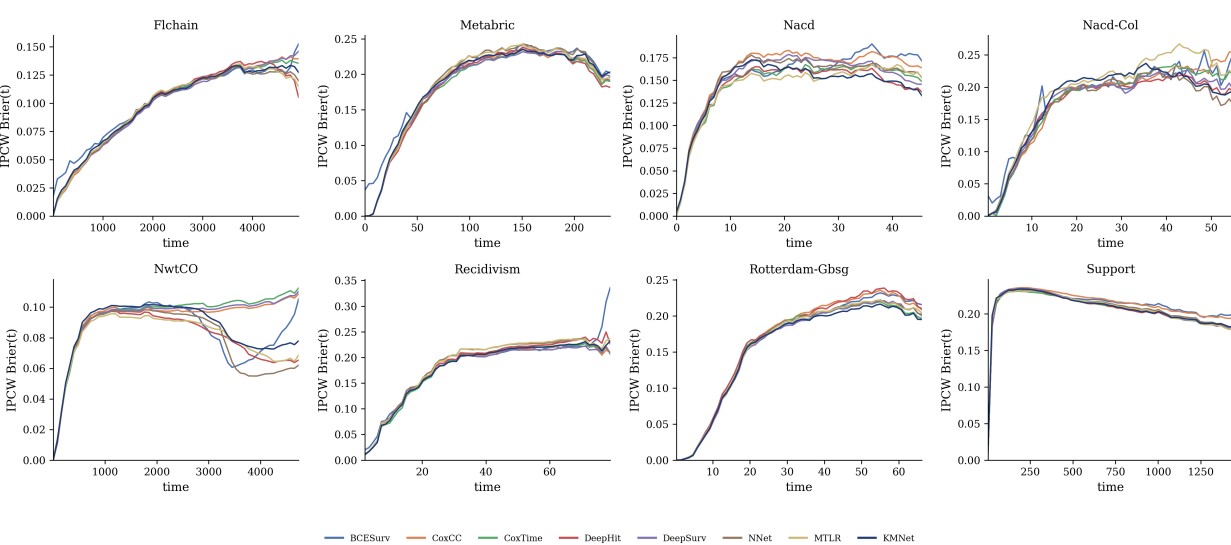

Figure 18: Time-dependent IPCW Brier score Brier($t$) for all eight models on each dataset. *Lower is better.* The temporal profiles vary by dataset, with prediction error generally increasing during early follow-up and either stabilizing or changing near the evaluation boundary. (prediction is hardest in mid-follow-up). Curves largely overlap, indicating comparable accuracy across models.

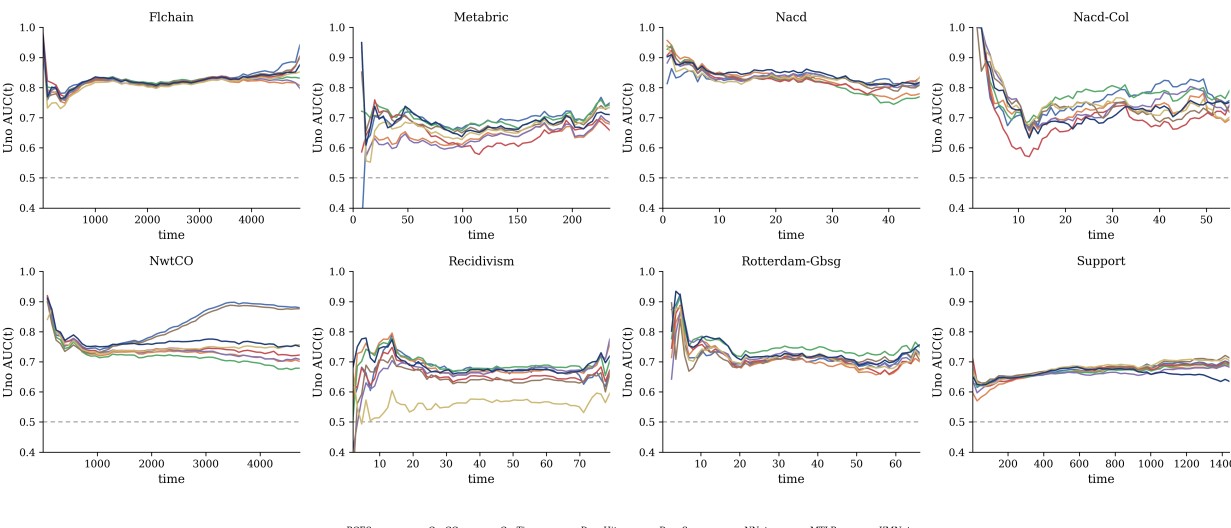

Figure 19: Time-dependent Uno cumulative/dynamic AUC AUC(*t*) for all eight models on each dataset. Higher is better; the dashed line at 0.5 is chance. AUC(*t*) measures discrimination (ranking)

and Nacd exhibit relatively stable AUC throughout follow-up, whereas Nwtco displays marked late-time separation among methods, Recidivism shows distinctly weaker AUC for MTLR over much of the horizon, and KMNet declines at later times on Support while most competing curves remain comparatively stable. These diagnostics therefore reinforce that relative model performance depends on both the dataset and the evaluation time, and that aggregate concordance and IBS values can obscure localized differences.

