# OpenReview forum: "Deep Discrete-Time Survival Analysis with Guaranteed Monotonicity"
_TMLR — Under review for TMLR_

### Review · Reviewer_X5GG · 2026-06-02

**Summary Of Contributions:**

Traditional discrete-time neural survival models trained with binary cross-entropy directly predict survival at each time point can yield non-monotone survival curves, which contradicts the interpretation of survival probability. To fill the gap, this paper propose Kaplan–Meier Net (KMNet) to predict interval-wise conditional survival probabilities and constructs the survival curve through cumulative multiplication, which enforces monotone non-increasing.
## Strength
1. KMNet predict interval-wise conditional survival probabilities, which guarantees monotonicity by design without sacrificing the representational flexibility of neural networks.
2. The design of weight term $w\_{ij}$ restricts supervision to time points at which an individual remains under observation, which can not be achieved in traditional Kaplan-Meier survival probability.
3. The KMNet extends traditional Kaplan-Meier survival probability estimation from two perspectives: (1) It considers the covariates so that achieve individual-level rather than group-level estimation. (2) It introduces a weight term to filter right censoring cases.

## Weakness
1. There are some repeated statements. For example, the content in Section 2.3 has already been introduced extensively in the introduction section.
2. The definitions and equations in Section 3 Kaplan–Meier Net are a little bit messy, a clear pipline figure would help.
3. Is it necessary to use 2 hyperparameter $\alpha$ and $\lambda$  to control $L_{Rank}$? More explanation is needed.

**Additional Comments:**

N/A

**Audience:**

Yes

**Audience Explanation:**

discrete-time survival model serves as an important topic in time-event related research, which influences the downstream decision making.

**Claims And Evidence:**

Yes

**Claims Explanation:**

Figrue 1 clearly demonstrates the drawback of discrete-time survival models trained with binary cross-entropy, which motivates the design of interval-wise conditional survival probabilities.

**Requested Changes:**

1. Fix the typo. For example: "In a typical time-to-event In studies"
2. Fig. 1b does not mean the phenomenon of predictions exhibit substantial increases over time is non-negligible, it just shows the distribution of upward jump amplitude.
3. All the problems shown in the Weakness part.

---

### Review · Reviewer_uAWp · 2026-06-24

**Summary Of Contributions:**

This paper studies discrete-time neural survival analysis. It points out that BCE-style models such as BCESurv, which directly predict marginal survival probabilities at each grid time, can produce non-monotone patient-specific survival curves. The authors propose Kaplan-Meier Net (KMNet), which predicts interval-wise conditional survival probabilities and constructs the survival curve by a cumulative product, guaranteeing monotonicity. The model is trained with a censoring-aware masked BCE loss and an additional ranking loss that compares individuals using the conditional survival probability at the event interval of the anchor observation. The paper evaluates KMNet on eight benchmark datasets against seven neural survival baselines, using time-dependent concordance and integrated Brier score.



My main concern is the novelty and positioning of the central contribution. The cumulative-product construction is very close to existing discrete-time hazard models such as Nnet-survival [1] and LogisticHazard discussed in [2], where survival is obtained by multiplying interval survival terms, equivalently one minus the interval hazards. These methods already imply monotone survival curves. The paper cites NNet only briefly and mainly as a baseline, but does not clearly discuss this equivalence. Therefore, the monotonicity-by-construction claim is overstated. The more plausible contribution is the event-interval conditional ranking loss, but the current ablations do not isolate it convincingly.



Reference

[1] A Scalable Discrete-Time Survival Model for Neural Networks, 2019.

[2] Continuous and Discrete-Time Survival Prediction with Neural Networks, 2019.

**Audience:**

Yes

**Audience Explanation:**

Yes. TMLR readers working on survival analysis, medical machine learning, and probabilistic prediction would likely be interested in the problem of invalid non-monotone survival curves from marginal BCE models. The finding is practically relevant, and a simple monotone discrete-time neural survival model with competitive performance is useful.

**Broader Impact Concerns:**

The paper targets patient-specific survival prediction, which can affect clinical decision-making. I do not see a specific ethical concern that would preclude publication, but the paper should avoid equating monotone survival curves with clinically reliable predictions.

**Claims And Evidence:**

No

**Claims Explanation:**

The empirical evidence supports the weaker claim that KMNet is a competitive discrete-time neural survival model. It achieves the best average rank for concordance and IBS and consistently improves over BCESurv. The statistical analysis is also useful.

However, the stronger methodological claims are not fully supported. KMNet outputs $p_j(x) = P(T > t_j | T > t_{j-1}, x)$. Existing discrete-time hazard models, including Nnet-survival and the LogisticHazard formulation, use $h_j(x) = P(t_{j-1} < T <= t_j | T > t_{j-1}, x)$, or equivalently the interval survival term $1 - h_j(x)$. Thus KMNet's $p_j(x)$ corresponds to $1 - h_j(x)$, and both approaches construct survival curves by cumulative products. Monotonicity by construction is therefore not new relative to this prior work.

The paper also lacks a sufficient related-work discussion. It does not mention NNet is only briefly described as a baseline. This makes the novelty claim unclear. The proposed ranking loss is interesting, but the paper does not provide the most important ablations, such as KMNet without ranking, KMNet with a global/DeepHit-style ranking loss, or LogisticHazard/NNet with the proposed ranking loss.

Finally, the empirical conclusions should be tempered. KMNet is not significantly better than several strong baselines under the reported tests. The results support "competitive with best average rank" more than broad superiority.

**Requested Changes:**

1. Add a substantive related work discussion of Nnet-survival, LogisticHazard/pycox, MTLR, PMF-based models, and DeepHit. The paper should explicitly state which prior models already guarantee monotone survival curves.
2. Reframe the novelty claim. The cumulative-product construction should not be presented as if it were absent from prior discrete-time neural survival models. The authors should clarify the relationship between KMNet's conditional survival probabilities and the interval hazards used in Nnet-survival / LogisticHazard.
3. Add ablations that isolate the ranking loss. At minimum, include KMNet without ranking, KMNet with a global/DeepHit-style ranking loss, and ideally LogisticHazard/NNet with the proposed ranking loss under the same backbone, grid, tuning budget, and splits.
4. Strengthen the comparison to NNet / LogisticHazard. Since these are the closest prior methods, the paper should explain whether they already produce monotone curves and compare their objective to KMNet's masked BCE formulation.
5. Temper the empirical claims. The manuscript should state that KMNet has the best average rank and improves consistently over BCESurv, but is not significantly better than several strong baselines.

---

### Review · Reviewer_aZ6P · 2026-07-05

**Summary Of Contributions:**

The authors propose a discrete-time neural survival model called KMNet. For each interval $j$ it predicts a conditional survival probability $p_j(x) = \sigma(\phi_j(x))$, and forms the patient-specific survival curve as the cumulative product $S(t_j \mid x) = \prod_{k \le j} p_k(x)$. Every factor lies in $[0,1]$, so the curve is non-increasing by construction (Proposition 1) and needs no post-hoc projection. Training combines a censoring-aware weighted BCE loss with a ranking term computed on the conditional probability at the anchor's event interval. The evaluation spans eight right-censored benchmarks against seven neural baselines, with TPE hyperparameter optimization, Friedman/Nemenyi and paired Wilcoxon tests, an objective ablation, and a simulation study.

Two contributions hold up: a survival head that returns mathematically valid monotone curves without projection, and an interval-local conditional ranking loss that ranks on the per-interval conditional probability at the anchor's event interval, where DeepHit instead ranks on a global risk score or cumulative quantity.


**Key strengths.**

1. Proposition 1 is correctly proven. A product of per-interval sigmoid outputs in $[0,1]$ is non-increasing, so the curves are valid by construction.
2. The statistical reporting is honest, including results that cut against the paper's own framing. Section 5 states that most baselines fall within the Nemenyi critical distance of KMNet, the IBS comparison with NNet is $p=1.0$, and the DeepSurv concordance edge is not significant after FDR correction.
3. The empirical protocol is broad: eight datasets, seven established baselines, TPE tuning, an objective ablation, a simulation study, and released code.
4. The interval-local conditional ranking loss is well specified, and it is the component whose novelty is most defensible.

**Key weaknesses**
Three headline claims are calibrated above the paper's own evidence, including broad performance superiority, the attribution of the discrimination gains to the ranking term, and clinical/calibration usefulness. Separately, monotonicity-by-construction is over-positioned relative to standard discrete-hazard models, and the experimental protocol is under-specified for a fair-comparison check.

**Audience:**

Yes

**Audience Explanation:**

A discrete-time survival model that guarantees valid curves on average rank across benchmarks is of interest to the survival-analysis and clinical-ML segments of the TMLR audience.

**Broader Impact Concerns:**

The submission motivates the method with clinical decision support and evaluates on a re-incarceration dataset, both high-stakes settings, and it carries no Broader Impact Statement. I recommend adding one that covers the risk of clinical over-interpretation of a retrospective-benchmark model, dataset provenance and consent for the medical and criminal-justice cohorts, and the prospective validation a real deployment would require.

**Claims And Evidence:**

No

**Claims Explanation:**

The core technical claim, valid monotone curves by construction (Proposition 1), is fully supported, and the best-average-rank result (1.62 concordance, 2.38 IBS) is supported by Section 5. Three headline claims are not.

First, the "best overall", "strong gains", and "strong discrimination and calibration" framing contradicts the paper's own significance tests, where KMNet is statistically indistinguishable from CoxTime, DeepHit, and DeepSurv, and is IBS-tied with NNet at $p=1.0$.

Second, the claim that the conditional ranking term drives the discrimination gains is untested. All eight ablation variants in Table 5 retain that term; there is no $\lambda=0$ (no-ranking) row and no global-CDF-ranking row, although Appendix A.1 describes the global-CDF variant.

Third, the clinical, calibration, and decision-support claims outrun the evidence. Calibration is summarized only by IBS, there is no decision-curve or D-calibration analysis. Also, the motivation median survival time is not measured, and real-cohort risk stratification is absent.

**Requested Changes:**

1. Should revise relevant parts (e.g., abstract, introduction) to ensure the performance claims match Section 5.

2. Add a $\lambda=0$ (no-ranking) KMNet control and the global-CDF-ranking control to the ablation to isolate the conditional ranking term, or remove the claims that attribute the discrimination gains to that term.

3. Add real-data calibration and decision evidence (per-horizon reliability, D-calibration, calibration slope/intercept, decision-curve analysis, real-cohort risk-stratification curves)

4. Scope the monotonicity contribution to the marginal-BCE (BCESurv) setting, state which compared baselines already guarantee monotone curves (six of the seven: NNet, MTLR, DeepHit, and the three Cox variants), and attribute the product-of-conditionals construction to the discrete-hazard literature.

5. Report the number of splits/seeds, the TPE trial budget $B$ (Algorithm 1), the IBS horizon $\tau$, the evaluation grid, and the IPCW details, and confirm a matched tuning budget and a matched evaluation horizon/grid across continuous- and discrete-time baselines.

6. Add a monotone post-processed BCESurv baseline (isotonic or cumulative-min), otherwise, end-to-end monotonicity is an alternative to post-hoc projection.

7. Formalize the contrast with DeepHit-style global/cumulative ranking by writing the comparator objective and contrasting it with $p_{\kappa_i}$.